# Triplet maintenance therapy of olaparib, pembrolizumab and bevacizumab in women with *BRCA* wild-type, platinum-sensitive recurrent ovarian cancer: the multicenter, single-arm phase II study OPEB-01/ APGOT-OV4

Yoo-Na Kim[1], Boram Park [2], Jae Weon Kim[3], Byoung Gie Kim[4], Sang Wun Kim [1], Hee Seung Kim[3], Chel Hun Choi[4], Myong Cheol Lim[5], Natalie YL Ngoi[6,7], David SP Tan[6,7,8,9] & Jung-Yun Lee [1]✉

In this multicenter, open-label, single-arm, Phase II study with Simon two-stage optimum design (NCT04361370), we investigate the efficacy and safety of triplet maintenance (olaparib, pembrolizumab, bevacizumab) in patients with platinum-sensitive recurrent ovarian cancer who are wild-type for *BRCA* 1/2. A total of 44 patients were enrolled, and the median follow-up duration was 22.9 months (interquartile range: 17.4–24.7). The primary outcome was 6-months progression-free survival (PFS), which was 88.6% (95% confidence interval [CI] 75.4–96.2), meeting the pre-specified primary endpoint. The secondary outcomes reported here include median PFS, 12-months PFS, and overall survival and safety. The median PFS was 22.4 months (20.4–∞), with a 12-months PFS rate of 84.0% (95% CI 69.3–92.0). The median overall survival was 28.6 months (27.3–∞). The combination demonstrated tolerable toxicity with manageable side effects. Other secondary outcomes include time-to-progression, time to subsequent treatment, time to second treatment and PFS2; however, this data is not reported, as treatment is still ongoing in a majority of patients. Exploratory analysis shows that patients who were homologous recombination deficiency-positive or had a programmed death-ligand 1 combined positive score ≥1 showed a favorable response (*P* = 0.043 and *P* < 0.001, respectively). Thus, triplet maintenance shows durable efficacy with tolerable safety in patients with platinum-sensitive recurrence.

Patients with ovarian cancer who have received primary surgery followed by platinum-based chemotherapy will most likely experience disease recurrence[1]. Once relapsed, patients inevitably follow the relentless disease trajectory hallmarked by increased resistance to therapy and shortened time to recurrence. The treatment for ovarian cancer is determined based on the treatment-free interval since the last platinum agent, and accordingly, patients are classified as having platinum-sensitive (relapse ≥6 months) or platinum-resistant (relapse <6 months) disease[2]. The standard of care for patients with platinum-sensitive recurrence is platinum-based chemotherapy[3]. However,

repeated exposure to platinum agents causes toxicity and, ultimately, therapy resistance.

In the platinum-sensitive recurrent cancer setting, maintenance with poly(ADP-ribose) polymerase (PARP) inhibitors was found to significantly improve progression-free survival (PFS) regardless of the BRCA mutation status[4–6]; this has led to PARP inhibitors being approved by the health regulatory agencies in the US[7], Europe[8], China[9], and Korea[10]. However, across all studies, their greatest benefit was reported in patients with BRCA mutations, with limited activity observed in BRCA wild-type patients[11]. Another approved maintenance option for platinum-sensitive recurrence is bevacizumab, an anti-angiogenic agent. However, the median PFS gain from adding bevacizumab was 3.4 months in GOG-213[12] and 4.0 months in the OCEANS trial[13]. Outcomes from these historical trials suggest that the use of antiangiogenic agents as monotherapy may be insufficient for recurrent disease. Therefore, studies to identify optimal treatments for BRCA wild-type patients with platinum-sensitive recurrent ovarian cancer are required.

To improve the outcomes for BRCA wild-type patients with ovarian cancer, various PARP inhibitor-based combinations have been suggested. The first is olaparib plus an antiangiogenic agent. The combination of olaparib plus cediranib showed an improved outcome in BRCA-wild-type patients with platinum-sensitive recurrent ovarian cancer when compared to olaparib alone; this may have been because cediranib led to the induction of homologous recombination deficiency (HRD)[14]. Furthermore, in the frontline maintenance setting, patients receiving maintenance with olaparib plus bevacizumab showed a significant PFS benefit compared to bevacizumab alone in BRCA-wild-type, HRD-positive patients, thus expanding the potential pool of beneficiaries for olaparib[15]. Another potential PARP inhibitor-based combination is olaparib with an immune checkpoint inhibitor (ICI), such as the anti-PD-L1 or anti-PD-1 agents, and the combination of durvalumab and olaparib has shown promising activity with manageable toxicity in recurrent ovarian cancer[16,17].

The aforementioned clinical studies, along with scientific research on the mechanisms[17,18], suggest that combining PARP inhibitor with an ICI and antiangiogenic agent in the maintenance setting may enhance the efficacy of PARP inhibitor monotherapy in BRCA wild-type patients with ovarian cancer. Several ongoing phase III trials, namely DUO-O (NCT03737643), KEYLYNK-001 (NCT03740165), and FIRST (NCT03602859), are exploring the triplet combination as maintenance therapy in a frontline setting. In this trial, we evaluated the efficacy and safety of triplet maintenance therapy in BRCA wild-type patients with platinum-sensitive recurrent ovarian cancer.

## Results

### Study design, enrollment, and patient demographics
Between October 20, 2020, and March 22, 2022, 44 patients were enrolled in the study and treated accordingly (Fig. 1); their baseline characteristics are shown in Table 1. The median age was 61 (range 43–78). Twelve patients (27.3%) progressed 6–12 months after their penultimate platinum therapy, and 33 (75.0%) showed a partial response (PR) after their most recent platinum therapy. In terms of biomarkers, 54.6% were HRD-positive (genomic instability score ≥42), and 63.6% had programmed death ligand-1 (PD-L1) combined positive score (CPS) ≥1. One patient received a PARP inhibitor, and 9 received bevacizumab as maintenance after first-line chemotherapy. Efficacy and safety analyses were completed for all 44 patients who received at least one dose of the study medication. At the data cutoff, 23 patients were still receiving treatment. Twenty-one patients discontinued treatment, including 17 patients with progressive disease (PD), 2 patients who completed the 2 years of treatment, 1 patient with myelodysplastic syndrome (MDS), and 1 patient who withdrew consent. The median follow-up duration was 22.9 months (interquartile range (IQR): 17.4–24.7).

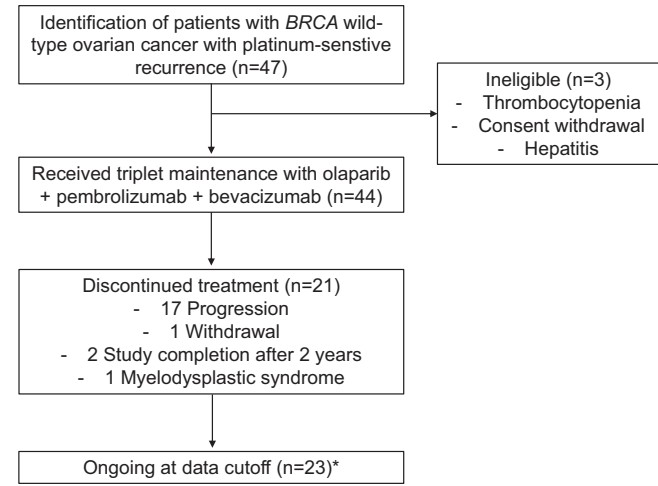

**Fig. 1 | Trial profile.** The number of patients included in the analysis.

### Efficacy
The study met the pre-specified primary endpoint, with a 6-month PFS rate of 88.6% (95% confidence interval (CI) 75.4–96.2). At the data cutoff point, 19 patients showed disease progression after a median of 13.7 months (IQR 8.6–20.8). Secondary endpoints were also investigated. Overall, the median PFS was 22.4 months (20.4–∞) (Fig. 2a). The 12-month PFS rate was 84.0% (95% CI 69.3–92.0), and 18-months PFS rate was 71.4% (95% CI 54.9–82.7%). An overall survival (OS) event occurred in 10 patients, which included two patients with treatment-unrelated deaths. One patient died of post-operative complications after undergoing surgery for a primary brain tumor; another patient died due to complications during the subsequent line of chemotherapy. The median OS was 28.6 months (27.3–∞) (Fig. 2b). Since a majority of patients were still ongoing at the data cutoff, other secondary endpoints such as time to progression, time to subsequent treatment, time to second treatment, and PFS2 were not reported.

The treatment overview for each patient, including the first platinum-free interval and duration of triplet maintenance therapy, is shown in Fig. 3. Patients are ordered in terms of decreasing duration from the start of first-line chemotherapy to the start of triplet maintenance therapy; the 6-month time point is marked with a vertical dashed line. For first-line therapy, nine patients and one patient had received bevacizumab and olaparib, respectively, as maintenance. Five of the 19 patients with PD showed disease progression within six months. One patient was determined to have progression after 4 months of triplet maintenance; however, therapy was continued at the clinician's discretion, and the treatment was ongoing at the data cutoff point.

### Safety and tolerability
All patients experienced at least one adverse event (AE) of any grade. The summary statistics for AE are shown in Supplementary Table 1. The most common AEs were nausea (59.1%), dyspepsia (56.8%), proteinuria (43.2%), general weakness (40.9%), anemia (38.6%), and neutropenia (38.6%) (Supplementary Table 2). Twenty-three (52.3%) of the 44 patients experienced grade 3 AEs, the most common of which was anemia (22.7%). One notable grade 3 event was small bowel perforation, which occurred in one patient after seven cycles of triplet maintenance therapy. At the time of the event, the small bowel perforation was determined to be probably related to bevacizumab. This patient was conservatively managed with antibiotics, and after 3 weeks, was found with PD and small bowel obstruction. There was one grade 4 AE

## Table 1 | Patient characteristics

| | Patients (n = 44) |
|---|---|
| Age, year (median, range) | 61 (43–78) |
| BMI, kg/m² (median, range) | 22.9 (16.7–30.1) |
| Histology subtype | |
| High-grade serous carcinoma | 41 (93.2%) |
| Low-grade serous carcinoma | 1 (2.3%) |
| Clear cell carcinoma | 1 (2.3%) |
| Endometrioid carcinoma | 1 (2.3%) |
| FIGO stage at diagnosis | |
| I or II | 6 (13.6%) |
| III or IV | 38 (86.4%) |
| Time to progression after penultimate platinum therapy | |
| 6–12 months | 12 (27.3%) |
| 12–24 months | 21 (47.7%) |
| 24+ months | 11 (25.0%) |
| Best response to most recent platinum therapy | |
| CR | 11 (25.0%) |
| PR | 33 (75.0%) |
| Maintenance after first-line chemotherapy | |
| Bevacizumab | 9 (20.5%) |
| Olaparib | 1 (2.3%) |
| HRD score (genomic instability score) | |
| <42 | 18 (40.9%) |
| ≥42 | 24 (54.6%) |
| Missing | 2 (4.5%) |
| PD-L1 CPS | |
| <1 | 15 (34.1%) |
| ≥1 | 28 (63.6%) |
| Missing | 1 (2.3%) |

*BMI* Body mass index, *FIGO* International Federation of Gynecology and Obstetrics, *CR* Complete response, *PR* Partial response, *HRD* Homologous recombination deficiency, *PD-L1 CPS* Programmed death ligand-1 combined positive score.

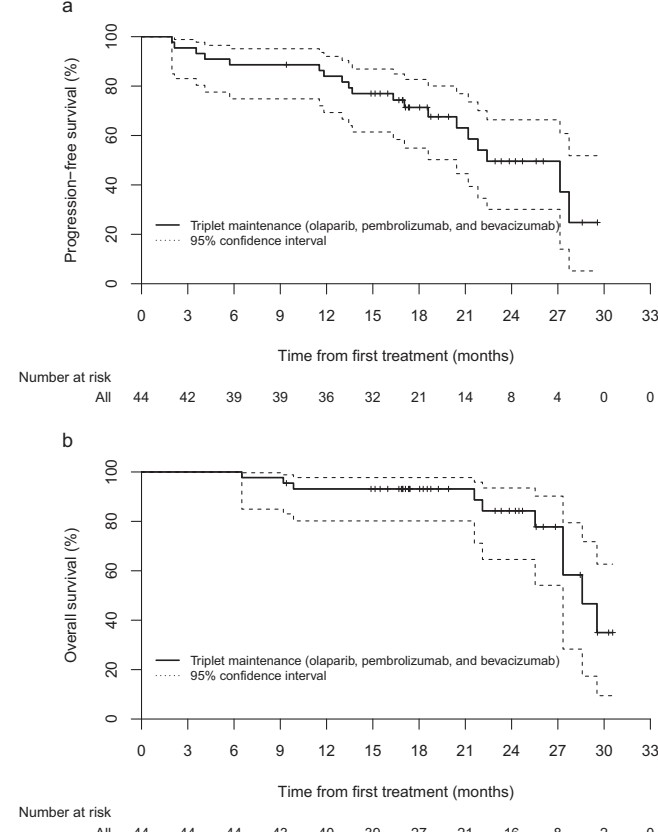

**Fig. 2 | Patient outcome. a** Progression-free survival and **b** overall survival at data cutoff. Source data are provided as a Source Data File.

where a patient developed MDS after 1 year of study maintenance. This patient was discontinued from the study treatment yet is disease-free at the data cutoff.

Twenty-seven (61.4%) of the 44 patients required a dose reduction for olaparib owing to an AE (general weakness [N = 8], anemia [N = 7], dyspepsia [N = 6], and nausea [N = 5]). With respect to each drug, dose interruptions were required in 38 patients (86.4%) for any of the three drugs, and in 32 patients (72.7%) for olaparib, 34 patients (77.3%) for pembrolizumab, and 33 patients (75.0%) for bevacizumab. Four patients permanently discontinued taking bevacizumab due to side effects (allergic rhinitis [N = 2], dyspepsia [N = 1], and general weakness [N = 1]) and continued the study with pembrolizumab and olaparib as per the study protocol.

Immune-mediated AEs were reported in 36 (81.8%) of the 44 patients. The most frequent immune-related AEs that were causally associated with pembrolizumab were thyroiditis [N = 9], blood thyroid stimulating hormone increase [N = 7], arthralgia [N = 6], aspartate aminotransferase increase [N = 6], fatigue [N = 6], and hyperthyroidism [N = 6]. Other notable immune-mediate AEs were diabetes mellitus [N = 1] and hypophysitis [N = 1], which were grade 3 and grade 2, respectively. Seven (15.9%) of the 44 patients experienced grade 3 immune-related AEs, including alanine aminotransferase increase [N = 1], blood thyroid stimulating hormone increase [N = 1], cellulitis [N = 1], diabetes mellitus [N = 1], an abnormal liver function test [N = 1], myalgia [N = 1], and rash [N = 1], and shingles [N = 1]. No grade 4 immune-mediated AEs were observed.

Overall, there were no newly identified AEs or immune-related AEs, aside from the type and frequency of events that could be expected from each agent based on previous reports. All events were managed conservatively and appropriately. Aside from one patient with MDS, there was no case of discontinuation from the study owing to AEs or treatment-related deaths.

### Exploratory outcomes

As exploratory outcomes, stratification was performed according to the pre-specified biomarkers (Supplementary Fig. 1). Patients with HRD-positive status showed improved PFS when compared to HRD-negative ($P = 0.043$); those with a PD-L1 CPS ≥1 showed improved PFS when compared to those with a PD-L1 CPS < 1 ($P < 0.001$). No significant difference was found regarding the response after second-line chemotherapy. A treatment overview plot stratified according to PD-L1 and HRD status is shown in Supplementary Fig. 2.

## Discussion

The OPEB-01 study investigated triplet maintenance with olaparib, pembrolizumab, and bevacizumab in *BRCA* wild-type patients with platinum-sensitive recurrent ovarian cancer. The study met the primary endpoint with a 6-month PFS rate of 88.6%. The response was durable, as supported by the efficacy data as secondary outcomes, which showed a median PFS of 22.4 months (20.4–∞) and a 12-months PFS of 84.0% (95% CI 69.3–92.0). The safety profile for the triplet combination was consistent with the known safety profiles expected for each agent individually.

The recently presented MEDIOLA study showed the promising efficacy of a triplet combination (olaparib, durvalumab, and bevacizumab) as a treatment strategy for germline *BRCA* wild-type platinum-sensitive recurrent ovarian cancer, with a median PFS of

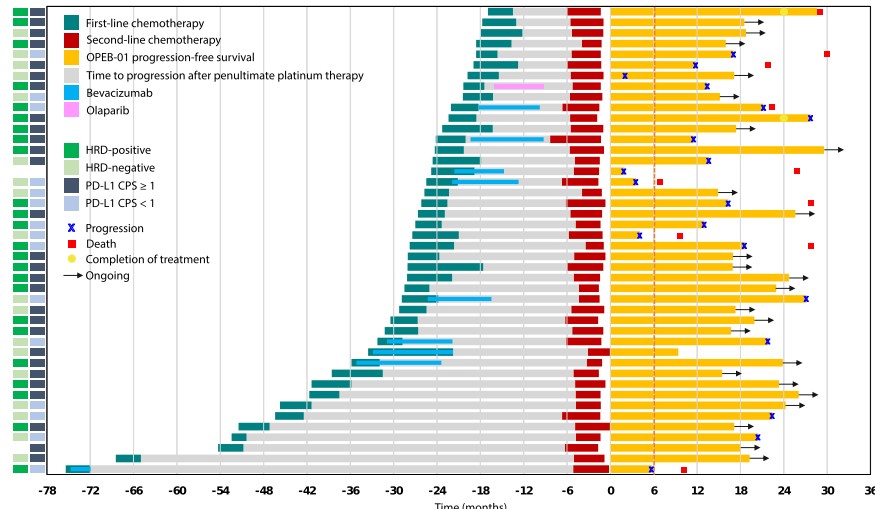

**Fig. 3 | Therapy outcomes showing first-line chemotherapy duration, platinum-free interval, and second-line chemotherapy duration, followed by triplet maintenance therapy.** Patients who are included in the ongoing triplet maintenance trial are marked with arrows; progression and death dates are marked. The 6 months time point since the start of triplet maintenance is marked with a vertical dashed line. HRD Homologous recombination deficiency, PD-L1 CPS Programmed death ligand-1 combined positive score. Source data are provided as a Source Data file.

15 months[19]. The most pronounced difference was that a triplet combination was used as a treatment in the MEDIOLA study and as maintenance in our study. Another difference was that the MEDIOLA study screened for patients based on germline *BRCA* status, whereas our study fully screened for both germline and somatic *BRCA*. In this study that exclusively included *BRCA* wild-type patients with platinum-sensitive recurrent ovarian cancer, the median PFS was 22.4 months. However, further maturation of the PFS data is necessary to elucidate the magnitude of benefit in maintenance versus treatment setting.

The efficacy of OPEB-01 can be compared to previous studies on currently available monotherapy options, namely PARP inhibitor and bevacizumab maintenance trials involving *BRCA* wild-type, platinum-sensitive recurrent ovarian cancer. In the OPINION trial, which investigated olaparib maintenance monotherapy in 279 patients without the germline *BRCA* mutation, the median PFS was 9.2 months (95% CI 7.6–10.9)[11]. There are two randomized trials involving PARP inhibitor monotherapy maintenance in *BRCA* wild-type patients: Study 19[20] for olaparib and NOVA[5] for niraparib maintenance in platinum-sensitive recurrent disease[5]. In the placebo groups of these two studies, the median PFS was consistently less than 6 months: 5.5 months for Study 19 and 3.9 months for NOVA. In comparison, in the PARP inhibitor maintenance subgroup, the median PFS was 7.4 months in Study 19 and 9.3 months in NOVA, translating into an absolute benefit of 1.9 months (HR 0.54, 95% CI 0.34–0.85) in Study 19 and 5.4 months (HR 0.45, 95% CI 0.34–0.61) in NOVA for PARP inhibitor compared to placebo. Furthermore, based on the PFS curves of these trials, the 12-months PFS rates were approximately 30% in the PARP inhibitor monotherapy group and 10% in the placebo group in these two trials. These findings are in contrast with the results from the OPEB-01 study, where the 12-month PFS rate was 84.0%. Overall, compared to monotherapy or doublet trials, the outcomes of our study suggest a potential synergy among the three different agents with an extension of the median PFS in a recurrent *BRCA* wild-type cohort beyond the benchmark of 19.1 months for patients with germline *BRCA* mutations in the SOLO-2 trial[21].

Furthermore, our efficacy outcome surpassed the median PFS of 18.9 months in the somatic *BRCA* wild-type subgroup of the PAOLA-1 study, which was a frontline maintenance study with doublet regimen (olaparib and bevacizumab)[15]. DUO-O, a randomized, placebo-controlled phase III trial, showed a significant improvement in PFS with first-line chemotherapy with durvalumab and bevacizumab,

followed by maintenance durvalumab, bevacizumab, and olaparib compared with control in patients with *BRCA* wild-type ovarian cancer[22]. The median PFS in DUO-O in the triplet maintenance arm was 24.2 months from the randomization. Direct comparisons between DUO-O and our study need to be interpreted with caution due to the differences in study design and the line of therapy. However, as shown in our study, the DUO-O study showed the efficacy of the triplet combination.

The toxicity profile in our study was in line with that of previous studies. The most common AEs were hematologic toxicities, including anemia (any grade 38.6%; grade ≥ 3 22.7%) and neutropenia (any grade 38.6%; grade ≥ 3 6.8%). Both the toxicity rate and profile were similar to those in previous studies on olaparib monotherapy (anemia of any grade 16.9–46.0%; grade ≥ 3 5.1–21.0%; neutropenia of any grade 15.8–24.0%; grade ≥ 3 1.8–8.0%)[11,20,21]. Although the rate of immune-mediated AEs (81.8%) was higher in our study than the reported rate of 22.6% in the Keynote 100 study[23], the events were mostly mild (grade 1 or 2). One of the most common immune-related AEs in our study was thyroiditis (20.5%), which was thyroid-related and thus similar to the most common AE in the Keynote 100 study, which was hypothyroidism (10.1%). Overall, the AEs and immune-related AEs were in line with those observed previously in the respective monotherapy studies, showing no evidence of drug–drug interactions among the three agents.

In terms of AE-related statistics, our study had high dose reduction and interruption rates, 61.4% and 86.4% (for any of the three study drugs), respectively. These rates were higher than those reported in previous studies on doublet regimen. For instance, our dose interruption rate of 86.4% surpassed 54% in PAOLA-1[15] or 65% in ATALANTE[24]. Similarly, our dose reduction rate of 61.4% was also higher than the 41% observed in PAOLA-1. There could be potential reasons. First, since all patients in our cohort were Asian, there could be ethnic differences. Second, we managed to achieve a low discontinuation rate through active dose reduction or interruption. In contrast, other studies frequently experienced discontinuation of the study drugs, such as 32.3% in MEDIOLA[19] with a median follow-up of 31.9 months and 26% in DUO-O[22] with a median follow-up of 23.3 months, whereas our study observed a discontinuation rate of 11.4%. Third, the triplet regimen could be associated with higher toxicity compared to mono or doublet regimen. However, the safety profile was generally consistent with that of the previous triplet

regimen (DUO-O)[22]. The rate of AEs leading to dose modification was 76% in DUO-O (dose interruption rate was not reported), and our AE profiles were also similar.

In terms of activity, previous clinical studies have suggested that a triplet combination (PARP inhibitor, ICI, and antiangiogenic agent) may be more effective than a doublet combination (PARP inhibitor and ICI), especially in *BRCA* wild-type patients. A previous phase II study with olaparib and durvalumab (anti-PD-L1) in *BRCA* wild-type patients with platinum-sensitive recurrence showed that VEGFR and PIGF expression was significantly increased in biopsy samples while the patients were receiving the PARP inhibitor[17]. Such compensatory increases in VEGF may lead to therapy resistance via decreased T-cell function and trafficking and increased PD-1 expression in CD8 T-cells[18]. Thus, adding antiangiogenic inhibitors may help relieve the potential cause of therapy resistance. The consistent activity of triplet combination across three studies, MEDIOLA[19], DUO-O[22], and our study, further supports this hypothesis.

In addition to improving the efficacy, our data have suggested that triplet maintenance therapy may help expand the potential target population beyond *BRCA* wild-type patients. Similar to the previous report from the PAOLA-1 study, our study observed longer PFS in patients with *BRCA* wild-type showing HRD tumors[10]. With respect to the PD-L1 status, our subgroup analysis suggested that patients with PD-L1 CPS ≥ 1 may benefit more from triplet maintenance than do those with PD-L1 CPS < 1, an observation that could be expected from the Keynote 100 study[23]. These are interesting aspects that could help form a hypothesis for large, phase III randomized trials.

Our study was limited by the fact that it was a single-arm, open-label study with a relatively small patient population and no comparator group. In terms of study design, we enrolled patients who had responded to second-line chemotherapy, making our cohort more favorable compared to previous studies where patients were enrolled regardless of their response to chemotherapy. Therefore, caution should be exercised when comparing our results with other maintenance trials, such as those involving bevacizumab, where the agent is administered concurrently with chemotherapy followed by maintenance, regardless of the response to chemotherapy. The 6-month PFS rate was chosen as the primary endpoint because this was a single-arm phase II study that evaluated signals for quick decision-making; based on previous randomized PARP inhibitor monotherapy trials, we expected that a majority of the patients would show recurrence within 6 months without maintenance therapy. However, it would be beneficial to have further survival maturation to determine whether the signals of durable responses translate into an overall survival benefit. Another limitation of our study is the small sample size, which was especially limiting for subgroup analysis of PFS concerning HRD or PD-L1 status. Additionally, we lacked an olaparib or bevacizumab monotherapy group as a comparator. Hence, a future randomized trial with triplet maintenance may be necessary. With these limitations in mind, the strength of our study is the homogenous patient population in a platinum-sensitive recurrent setting. All patients were screened for germline and somatic *BRCA* status prior to enrollment. Pre-specified biomarkers, including HRD and PD-L1 status, were also assessed in most patients.

In conclusion, findings from the OPEB-01 study show that the triplet maintenance therapy with olaparib, pembrolizumab, and bevacizumab leads to promising outcomes and is tolerable in *BRCA* wild-type patients with platinum-sensitive recurrent ovarian cancer. Further research on biomarkers such as tumor microenvironment and RNA sequencing in pre- and post-treatment biopsies will be necessary to assess the specific mechanism of response and identify the patient subsets that would benefit most from triplet maintenance therapy. The long-term outcomes of triplet maintenance therapy will need to be further explored with survival maturation and additional randomized studies.

## Methods

The trial was conducted in accordance with the Declaration of Helsinki and the Guidelines for Good Clinical Practice. The trial was approved by the institutional review board of each institution (Severance Hospital: 4-2020-0386; Seoul National University Hospital: H-2101-017-1186; Samsung Medical Center: SMC 2020-08-078; National Cancer Center: NCC2021-0069; National University Cancer Institute: 2020/01198). Written informed consent was obtained from all participants before study enrollment. Patients did not receive any compensation for their participation. The trial was registered under the name "Olaparib Maintenance With Pembrolizumab & Bevacizumab in *BRCA* Non-mutated Patients With Platinum-sensitive Recurrent Ovarian Cancer (OPEB-01)" (ClinicalTrials.gov identifier: NCT04361370) on April 2020.

### Study design and participants

OPEB-01/Asia-Pacific Gynecologic Oncology Trials Group (APGOT)-OV4 is an investigator-initiated, multicenter, single-arm, open-label, phase 2 study that was conducted in five medical centers across Korea and Singapore (Supplementary Table 3)[25]. The first patient was enrolled on October 22, 2020, and the last patient was enrolled on March 22, 2022. Eligible patients were women ≥ 20 years of age, with an Eastern Cooperative Oncology Group performance status of 0 or 1, histologically confirmed epithelial ovarian cancer, and lacking germline and/or tumor *BRCA* mutations. Gender was not considered in the study design since this trial was on women's cancer. With respect to histology, patients with high-grade predominantly serous, endometrioid, carcinosarcoma, mixed Mullerian with high-grade serous components, clear cell, or low-grade serous ovarian cancer, primary peritoneal cancer, or fallopian tubal cancer were considered. A cap of eight patients was applied for clear cell carcinoma; mucinous carcinoma could be enrolled. Patients had received two previous courses of platinum-containing therapy and showed platinum-sensitive disease (platinum-free interval of ≥6 months) following their penultimate platinum course, along with a complete response (CR) or PR to their most recent platinum course; they were enrolled in the study within eight weeks of completing their final platinum regimen, regardless of prior PARP inhibitor or bevacizumab use but had to be immunotherapy naïve. The full eligibility criteria are presented in the study protocol (Supplementary Note).

### Procedures

Patients received triplet maintenance therapy with olaparib (300 mg tablets, orally twice daily) and bevacizumab (15 mg/kg, intravenously), followed by a combination of 300 mg olaparib twice daily (up to two years and longer in case of PR at two years), 200 mg pembrolizumab every 3 weeks (cycles 2 through 35), and 15 mg/kg bevacizumab every 3 weeks intravenously until progression or intolerable toxicity. Unlike olaparib and bevacizumab, which were started in cycle 1, pembrolizumab was initiated in cycle 2, based on the preclinical rationale that PARP inhibitors induce immune cell infiltration and PD-L1 upregulation, leading to enhanced antitumor immunity that can be further enhanced through the combination of an immune checkpoint inhibitor. Patients were allowed to withdraw from the study at any time.

Dose modifications to manage toxicities were allowed. Olaparib toxicities were managed with supportive care, dose interruptions, or dose reductions (two lower dose levels were allowed: 250 mg twice daily and 200 mg twice daily). If a patient could not tolerate olaparib at 200 mg twice daily, the patient had to be discontinued. Dose re-escalation was also not permitted, but dose interruptions of less than 4 weeks were permitted. Hematotoxicity was monitored and managed as specified in the protocol (Supplementary Note). With respect to AE reporting, we have adhered to the exact terms used by clinicians. Pembrolizumab and bevacizumab toxicities could be managed with supportive care or dose interruptions; dose reductions were not permitted. Patients were discontinued if pembrolizumab was interrupted

for 12 weeks or longer due to AEs or toxicity or for ≥3 weeks due to administrative causes. Bevacizumab was considered a background therapy; its administration was based on the clinicians' discretion, and patients were allowed to continue with olaparib and pembrolizumab if bevacizumab was interrupted or discontinued. Prophylaxis for nausea and vomiting was not mandatory but was allowed. Tumor assessment was performed using computed tomography or magnetic resonance imaging of the chest, abdomen, and pelvis every three cycles for the first 2 years, every four cycles from the second to the third year, and every six cycles from the third year onward. Assessments were performed up to 7 days before or after the designated time point by the investigator using the Response Evaluation Criteria in Solid Tumours version 1.1[26].

For biomarker analysis, archival tumor tissues were collected from all patients. These biomarkers were pre-determined based on previous reports on monotherapy. For instance, PD-L1 was considered a biomarker for pembrolizumab based on the Keynote-100 study[23], and HRD status for olaparib was determined based on the PAOLA-1 study[15]. Immunohistochemistry (IHC) was performed using a Ventana Benchmark XT automated stainer (Ventana Medical Systems, Arizona, United States) with antibodies against PD-L1 (pre-diluted, clone 22C3, DAKO, Glostrup, Denmark). PD-L1 expression in the tumor cell membrane and the membrane and/or cytoplasm of tumor-associated mononuclear inflammatory cells was scored. The CPS was defined as the total number of tumors and immune cells stained with PD-L1 divided by the number of all viable tumor cells and then multiplied by 100. Genomic scarring was estimated by determining copy number alterations in the whole exome sequencing data using Sequenza-utils (v.3.0.0)[27], based on the loss of heterozygosity, large-scale transitions, and the number of telomeric allelic imbalances, and these were estimated using the scarHRD (R package v.0.1.1)[28]. The sum of these values served as the genomic scar score and was used as the input seqz file[29–31]. Based on the genomic scar score and a cutoff of 42, HRD status was determined.

### Outcomes
The primary endpoint was the 6-month PFS rate. PFS was defined as the time from the start of treatment to the first documented sign of disease progression or death from any cause. The reported secondary endpoints included PFS, OS, and safety. Other secondary endpoints, such as time time-to-progression, time to subsequent treatment, time to second treatment, and PFS2 were not reported because a majority of patients were still ongoing at data cutoff. OS was defined as the time from the first treatment to death from any cause. The cutoff date was May 25, 2023. Investigation of biomarkers of response was a pre-specified exploratory outcome.

### Statistical analysis
The study was conducted using Simon's two-stage optimal design with assumptions concerning the estimated PFS rate in ovarian cancer. As the benchmark for the null hypothesis, we chose the GOG-213 study, which investigated chemotherapy plus bevacizumab followed by bevacizumab maintenance regardless of BRCA mutations. Recognizing the conceivable differences between GOG-213 and our trial, which focuses on the maintenance therapy, we used the best approximation from GOG-213 by considering the chemotherapy time window because of the lack of data on studies with bevacizumab maintenance in patients responding to chemotherapy. Thus, based on the current standard of care and the best approximation from GOG-213, the rate of patients with a disease-free state at 6 months was expected to be 50% with bevacizumab maintenance. Moreover, the HR of adding maintenance therapy with a triplet combination (PARP inhibitor, ICI, and anti-angiogenic therapy) was assumed to be 0.5, equivalent to a PFS rate of 70.7%. The null hypothesis for this study would be a 6-month PFS

rate of 50%, and the alternative hypothesis of interest would be a 6-month PFS rate of 70%. Using Simon's two-stage optimal design at a one-sided 5% level of significance and 80% power, 39 patients were included in this study. In the first stage, 22 patients would be enrolled; if 10 or more PDs were observed, the trial would be terminated. Else, the trial would continue to the second stage. The null hypothesis would be rejected if the total number of PDs was less than 15. Considering loss to follow-up, the 44 patients would be studied.

The proportion of patients achieving responses and 95% CIs was assessed using the Clopper–Pearson exact method. Survival analyses were pre-specified as secondary endpoints. The PFS and associated 95% CIs were calculated using the Kaplan–Meier method. A log-rank test was used to compare the PFS between the patient subsets. Statistical analyses were performed using SAS (version 9.4; SAS Institute, Cary, NC, USA).

### Reporting summary
Further information on research design is available in the Nature Portfolio Reporting Summary linked to this article.

## Data availability
The full study protocol and statistical analysis plan are available in the Supplementary Note. Data underlying all figures are provided in the Source Data file. Further data are not publicly available due to patient privacy but can be accessed on request from the corresponding author J.Y.L. (jungyunlee@yuhs.ac), for 10 years; individual de-identified participant data will be shared for academic research purposes. Source data are provided with this paper.

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

## Acknowledgements

This study was an investigator-initiated trial. MSD supported the study by providing the study drugs (olaparib and pembrolizumab). This work was also supported by the Innovation Growth Engine for Planning and Demonstration of the Commercialization Promotion Agency for R&D Outcomes (NTIS-1711121252), funded by the Korean government (MSIT) (recipient: J.Y.L.); National Medical Research Council Singapore Clinician Scientist Award Senior Investigator Grant (CSASI21jun-003) and the Pangestu Family Foundation Gynaecological Cancer Research Fund (recipient: D.S.T.). Yonsei University provided the study sponsorship. The drug provider or funding source was not involved in the study design, data collection, analysis, or manuscript writing. We appreciate all the patients, their families, and all the contributors at each participating site. We would like to thank Petar Jelinic, Sarper Toker, and Soo Yeon Ihm from the MSD for helping to initiate this investigator-initiated trial. Collaborators who helped enroll patients for this study were Sunghoon Kim and Yong Jae Lee from Severance Hospital, Mi-Kyung Kim from Ewha Womans University Mokdong Hospital, Taek Sang Lee from Seoul National University Boramae Medical Center, Won Kyo Shin and Heon Jong Yoo from Choong Name Sejong Hospital, Kyung Jin Min from Korea University Ansan Hospital, Hyun Woon Cho from Korea University Kuro Hospital, and Sook-Hee Hong from Seoul Saint Mary's Hospital, the Catholic University of Korea. We would also like to acknowledge Eun Hae Cho from GC Genome and Jung Kyoon Choi from KAIST, who assisted with the biomarker study.

## Author contributions

J.Y.L. was responsible for the conception and design of the study. J.W.K., B.G.K., S.W.K., H.S.K., C.H.C., M.C.L., N.Y.N., D.S.T., and J.Y.L. enrolled patients and collected the data. B.P. and J.Y.L. were responsible for the methodology. Y.N.K., B.P., and J.Y.L. verified the raw data. Y.N.K., B.P., and J.Y.L. analyzed the data. Y.N.K., B.P., and J.Y.L. participated in the data interpretation and writing of the manuscript. J.W.K., B.G.K., N.Y.N., and D.S.T. were responsible for reviewing and editing of the manuscript.

## Competing interests

MSD supported the study by providing the study drugs (olaparib and pembrolizumab). J.Y.L., B.G.K., and J.W.K. received grants from the MSD during the conduct of the study. J.Y.L. received grants and personal fees from AstraZeneca, Beigene, Bergenbio, Clovis Oncology, Immunogen, Janssen, Merck, Novartis, Roche, Seagen, Synthon, and Takeda. B.G.K. received grants from AstraZeneca, Cellid, and Utilex. J.W.K. received personal fees from AstraZeneca, Janssen, Takeda, GSK, Boryung, CMIC, LG Pharma, and Vifor Pharma. D.S.T. reports personal fees for advisory board membership from AstraZeneca, Bayer, Boehringer Ingelheim, Eisai, Genmab, GSK, MSD, and Roche; personal fees as an invited

speaker from AstraZeneca, Eisai, GSK, Merck Serono, MSD, Roche, and Takeda; ownership of stocks/shares of Asian Microbiome Library (AMiLi); institutional research grants from AstraZeneca, Bayer, Karyopharm Therapeutics, and Roche; institutional funding as coordinating PI from AstraZeneca and Bergen Bio; institutional funding as local PI from Bayer, Byondis B.V. and Zeria Pharmaceutical Co Ltd; a previous non-renumerated role as Chair of the Asia- Pacific Gynecologic Oncology Trials Group (APGOT); a previous non-renumerated role as the Society President of the Gynecologic Cancer Group Singapore; non-renumerated membership of the Board of Directors of the GCIG; and product samples from AstraZeneca, Eisai, and MSD (non-financial interest). N.Y.N. received personal fees from AstraZeneca and Pfizer. Otherwise, the authors declare that they have no conflicts of interest.

## Additional information

[1]Department of Obstetrics and Gynecology, Yonsei University College of Medicine, Seoul, Korea. [2]Biomedical Statistics Center, Research Institute for Future Medicine, Samsung Medical Center, Seoul, Korea. [3]Department of Obstetrics and Gynecology, Seoul National University, Seoul, Korea. [4]Department of Obstetrics and Gynecology, Samsung Medical Center, Sungkyunkwan University School of Medicine, Seoul, Korea. [5]Gynecologic Cancer Branch & Center for Uterine Cancer, National Cancer Center, Goyang, Korea. [6]Department of Medicine, Yong Loo Lin School of Medicine, National University of Singapore, Singapore, Singapore. [7]Department of Haematology-Oncology, National University Cancer Institute, National University Hospital, Singapore, Singapore. [8]National University of Singapore (NUS) Centre for Cancer Research (N2CR), Yong Loo Lin School of Medicine, National University of Singapore, Singapore, Singapore. [9]Cancer Science Institute, National University of Singapore, Singapore, Singapore. ✉e-mail: JUNGYUNLEE@yuhs.ac

