## [Peer Review File · Nature Communications]

Reviewers' Comments:

Reviewer #1:

Remarks to the Author:

My thanks to the editors for allowing me to review this manuscript and authors for their lucid exposition of the survival results describing the effect of the combination maintenance strategy of olaparib (ola) + bevacizumab (bev) + pembrolizumab (pembro).

Also the authors explained very carefully the goal of such maintenance therapy and the benifice compared to that we have currently access in routine (only bev or ola)we have to be cautious about the current standard of care and the data from the litterature:

currently 2 options are available for patients with sensitive relapse can received

- platinum based chemotherapy plus bev followed by bev monotherapy whatever is ther response to the CT+BEV

or

- platinum based CT alone and for patients in PR or CR they can received a maintenance with parpi alone

so one of the major comment for this trial is the included population did not matched with what we used currently (bev before randomisation or no bev) and we can say the same for pembro who in general need to be started with CT before to be continue for maintenance (as seen in other tumor model as cervical cancer or endometrial....)

for a statistical point of view the hypothesis cannot be extrapolated from GOG213 or litterature as the patients in these publication have a median PFS integrating all patients (not only the good responders) and analysis of the PFS starts with CT.

the other topic well mentioned by the authors is the mono arm trial: we are not able to make any comparison from a potential control arm including bev alone or olaparib alone. The main limitation of the study remains lack of a control arm without olaparib, which is discussed, but is a considerable flaw of design.

the important conclusion is the triplete is safe and well tolareted by the patients with majority of grade 1 and 2. However, it will relevant for the reader to see the % of dose discontinuation individually and all by patients.

from the exploratory analysis, the subgroups of CPS1 seems encouraging and need to develop looking to what we are waiting for in the 1st line setting considering several phase III trial on going

minor aspects can be also discussed

- about nomenclature BRCA non mutated can be replace by BRCAwt.

- other comment figure 3, as there is no correlation between efficacy and previous therapy or PFI, i will be in favor to present the data by HRD score and PDL1 expression

- adding the median follow up can be help full to interpret the PFS.

Reviewer #2:

Remarks to the Author:

Key results:

This is a single arm phase 2, 44 patient study investigating the maintenance triplet therapy of Olaparib, Pembrolizumab and Bevacizumab in recurrent non-BRCA mutant platinum sensitive ovarian cancer following response to platinum-based chemotherapy. The study reports on the primary efficacy endpoint of PFS at 6 months and safety. The key findings are that this triplet maintenance therapy is safe and shows promising efficacy as evidenced by the PFS6 and perhaps the more meaningful secondary endpoints of PFS12 and median PFS not reached at data cut.

Validity

The use of the PFS6 as primary endpoint is questionable. PFS6 is not a validated endpoint recognised as a surrogate of overall PFS benefit. This is important here as especially given this group of patients have just responded well to chemotherapy so a high PFS6 rate would be expected. However the high PFS12 months is impressive and supports the authors claims of duration of efficacy.

The authors also recognise the limitations of a single arm study.

Significance

This is novel and the first triplet maintenance trial to report out so there is no conflicting published literature. It does align with other reports such as the MEDIOLA trial which has shown that this triplet non-chemotherapy sparing treatment is effective as a treatment in itself in the platinum sensitive setting, so the promising activity as maintenance treatment is not surprising to me as an expert in this field.

Data and methodology

Methodology -

I would question the inclusion/exclusion criteria in the trial and request more is added to the main manuscript for example:

1. Why include non-high grade serous histologies here? There is no data to support this and we know clear cell cancers and low grade serous are totally different cancers with different genetic drivers. The authors should justify this or not.
2. Why was Pembrolizumab not given in C1, what was the rationale for this?
3. The cycle length for pembrolizumab and bevacizumab are not included in the text and should be (they are in a figure only)
4. Which HRD biomarker testing was used?
5. What the requirement for previous chemo cycles ? at least 4 or must be 6?

Data itself is presented clearly and figures and tables easy to interpret. There is an error in Figure 1. its says BRCA mutant but should say 'NON-BRCA mutant'

Analytical approach

The authors should justify why they chose PFS6 as the primary endpoint given my previous comments as this would be an 'easy win'

There is limited analysis due to the small sample size but the association with PDL-1 CPS status is interesting and should be further explored.

Safety analysis - suggest only use CTCAE definitions in the table so avoid multiple terms for possibly the same thing - hyperthyroidism and thyroiditis fro example. More information should be provided in the main text about the 1 patient with the small bowel perforation as this is a concerning toxicity of the bevacizumab.

Suggested improvements

Suggested improvements detailed in the comments in the above sections

Also mention OCEANS phase 3 maintenance BEV trial as perhaps a better comparator than GOG213 as the later study also included surgery. (Oceans Trial Aghajanian C et al., JCO 2012)

Clarity and context

Well written and clear

Error in Figure 1 pointed out in the previous sections.

References

Yes appropriate references used but also add the OCEANS trial reference

Your expertise

Nothing outside of my scope although I do have statistical expertise.

Reviewer #3:

Remarks to the Author:

Dr. Kim and colleagues presented a multi-center, single-arm phase II study of triplet maintenance in BRCA non-mutated, platinum-sensitive recurrent ovarian cancer. The paper is well written, but some issues about the statistical analysis and swimmer plot need to be addressed.

1. The abstract needs more detailed information including the number of patients in the study, the study design (Simon's two stage).

2. Line 95: Please confirm that the median follow-up was estimated using the Kaplan-Meier method. If that was not the case, please modify accordingly. The IQR can also be estimated from the KM curve for follow-up using the 25th and 75th percentiles.
3. Line 99, 141: The progression is a time to event outcome, so the median time should be estimated through Kaplan-Meier method.
4. Line 100: Add text to show that this patient died after PD to make it clear
5. Line 102: Confidence interval of ORR should be reported given the statistical analysis part in method section
6. Line 288: Genomic scarring is defined here but not reported in the manuscript.
7. Line 298,319: Some secondary endpoints and median duration of response were not reported in the paper. It can be either removed or add those outcomes in the result section
8. Line 300-301: Censoring date for PFS should be based on last scan date for the patient since the endpoint is using progression per RECIST 1.1. Censoring for OS should be based on actual follow-up for each patient. These dates could be after the September 20, 2022 cut-off but then the correct analysis would be to use last scan date or September 20, 2022, whichever occurred first. Similarly for OS.
9. Fig 3:
 - a. There were 44 patients reported in this plot with 35 patients who are still receiving treatment. This is contradicted with the Fig 1 in which indicated that 33 patients are ongoing at the data cutoff.
 - b. Add the death mark to the plot
 - c. Add the first response mark so we can know duration of response
 - d. Move beginning of triplet maintenance therapy to $x=0$ as this is the primary endpoint of the study.
 - e. Add a vertical line at 6 months so primary endpoint (6-month PFS rate) can be read from the figure.

Dear Editor and reviewers,

Thank you for reviewing our manuscript and considering for publication. We have revised the manuscript based on the reviewers' comments. All revisions made to the manuscript are highlighted in the document. When we were reviewing the outcome plot (Fig. 3) as per Reviewer #3's comment, we discovered that one patient who voluntarily withdrew from the study was not reflected. Thus, outcomes have been updated throughout the manuscript (text, figures, and supplementary) to reflect this one patient. All authors have read and approved the revised version of the manuscript. Herein, point-by-point responses are presented. Responses are colored in blue, and changes to the original manuscript are highlighted in yellow.

Reviewer #1 - Ovarian cancer clinical trials (Remarks to the Author):

My thanks to the editors for allowing me to review this manuscript and authors for their lucid exposition of the survival results describing the effect of the combination maintenance strategy of olaparib (ola) + bevacizumab (bev) + pembrolizumab (pembro).

Also the authors explained very carefully the goal of such maintenance therapy and the benifice compared to that we have currently access in routine (only bev or ola)we have to be cautious about the current standard of care and the data from the litterature:

currently 2 options are available for patients with sensitive relapse can received

- platinum based chemotherapy plus bev followed by bev monotherapy whatever is ther response to the CT+BEV

Or

- platinum based CT alone and for patients in PR or CR they can received a maintenance with parpi alone so one of the major comment for this trial is the included population did not matched with what we used currently (bev before randomisation or no bev) and we can say the same for pembro who in general need to be started with CT before to be continue for maintenance (as seen in other tumor model as cervical cancer or endometrial....)

Q1) for a statistical point of view the hypothesis cannot be extrapolated from GOG213 or litterature as the patients in these publication have a median PFS integrating all patients (not only the good responders) and analysis of the PFS starts with CT.

A1) Thank you for your comment.

OPEB-01 includes patients who responded (CR/PR) to the last platinum-based chemotherapy, and we administered triplet maintenance with olaparib, pembrolizumab, and bevacizumab. On the other hand, in

GOG-213 patients were started with chemotherapy plus bevacizumab, followed by bevacizumab maintenance, and the median PFS was around 13.8 months with chemotherapy plus bevacizumab, counting from the start of chemotherapy. As the reviewer commented, the study population from GOG-213 included all patients from the start of chemotherapy, regardless of *BRCA* mutation. We acknowledge the differences in the study population between OPEB-01 and GOG-213.

Our sample size calculation was based on the estimated PFS rate in ovarian cancer. We assumed that the disease-free rate at 6 months is expected to be 50% with bevacizumab maintenance for the following reasons.

First, it would have been better if there was – as H0 or historical control – a study which have used bevacizumab maintenance in patients who have responded to 2L platinum-based chemotherapy. However, there is no such study so far. Therefore, we had no option but to perform statistical calculation based on the best approximation of the standard of care (GOG-213 on bevacizumab as maintenance). We have roughly calculated the median PFS with bevacizumab maintenance to be approximately 6 months from the end of chemotherapy plus bevacizumab, assuming a preceding chemotherapy period of 6-7 months (6 cycles plus approximately 4-8 weeks of pause before maintenance therapy).

Second, at the time of study design (year 2018), bevacizumab maintenance base on the result from GOG-213 was the only accessible maintenance option for *BRCA* wild-type patients with platinum sensitive recurrent ovarian cancer in Korea. For *BRCA* wild-type patients, there were only two options in 2L – either bevacizumab or no maintenance. Of the two options, we used bevacizumab maintenance as H0 because the outcome with bevacizumab was expected to be superior to giving no maintenance at all. As we specified in the study protocol (protocol page 56), applying the same expected efficacy HR of 0.5 (such as other triple maintenance therapy study, DUO-O), the alternative hypothesis (H1) would still be 70% of 6-month PFS rate. Had we used OCEANS trial as a comparator, from the start of maintenance with bevacizumab without chemotherapy, the median PFS would still be approximately 5-6 months.

Since 2021, the Korean FDA approved the use of PARPi monotherapy maintenance for patients with *BRCA* wild-type disease with platinum-sensitive recurrence. Thus, we have looked at the outcomes in this setting, where only the patients with CR/PR to the last platinum-based chemotherapy were enrolled. According to Study 19 (Ledermann et al., 2012) and NOVA trial (Mirza et al., 2016), the median PFS in *BRCA* wild-type patients was 7.4 months for olaparib and 9.3 months for niraparib. Furthermore, when we look at the median PFS in the control arm (i.e., no maintenance) of these two trials, the median PFS

was 5.5 months in Study 19 and 3.9 months in NOVA, both of which were consistently less than 6 months. We have added these aspects in the discussion section (please see the modifications at the end of A2).

In conclusion, while there are differences in the study population, the duration of bevacizumab maintenance post chemotherapy can be predicted as approximately 6 months. We performed statistical calculation based such approximation of the standard of care at the time of study design, yet there were limitations. Thus, we vigorously reviewed the PFS findings based on the previous trials (Study 19 or NOVA) with cohorts which bear more similarity with our study, and still found that in the control (i.e., no maintenance) or PARPi monotherapy arm the median PFS was around 6 months. Thus, we think it is admissible to have used the median PFS of 6 months as an approximation of PFS in H0. We have briefly mentioned this aspect in the methods section.

After revision (Methods, lines 373 – 384 in the marked manuscript): **As the benchmark for the null hypothesis, we chose the GOG 213 study, which investigated chemotherapy plus bevacizumab followed by bevacizumab maintenance regardless of *BRCA* mutations. Recognizing the conceivable differences between GOG 213 and our trial, which focuses on the maintenance therapy, we used the best approximation from GOG 213 by considering the chemotherapy time window, because of the lack of data on studies with bevacizumab maintenance in patients responding to chemotherapy. Thus, based on the current standard of care and the best approximation from GOG 213, the rate of patients with a disease-free state at 6 months was expected to be 50% with bevacizumab maintenance (current standard of care and results from GOG 213), and, Moreover, the HR of adding maintenance therapy with a triplet combination (PARP inhibitor, ICI, and antiangiogenic therapy) was assumed to be 0.5, which was equivalent to a PFS rate of 70.7 %.**

Q2) the other topic well mentioned by the authors is the mono arm trial: we are not able to make any comparison from a potential control arm including bev alone or olaparib alone. The main limitation of the study remains lack of a control arm without olaparib, which is discussed, but is a considerable flaw of design.

A2) We agree with you on the limitation of a single-arm trial design. Nonetheless, we think that an indirect comparison can still be made with the previously published trials involving PARPi monotherapy or bevacizumab in *BRCA* wild-type patients (please check the table below for the summary of outcomes). For PARPi monotherapy trials, the median PFS was 7.4 to 9.4 months. In these studies, the PFS6 ranged from 40-60% and PFS12 from 25-30% based on the best approximation from the PFS KM curve. For

bevacizumab trials, when we take into consideration the 6-7 months lead-time due to chemotherapy plus bevacizumab preceding the maintenance period, the median PFS from the start of maintenance was 5.4-6.8 months. Based on the best approximation from the PFS KM curve, from the start of chemotherapy the 12 months PFS was 50-60% and 18 months PFS was 35-45%, which would be equivalent to PFS6 and PFS12 when considering the 6-7 months lead-time.

Acknowledging the inherent limitation of a single arm phase II trial, we think that the series of outcomes in our study (median PFS, PFS6, and PFS12) are still notable. We have modified the discussion section to include a comprehensive comparison with previous studies, as well as the strength and limitation of our study.

Study type	Trial name	Median PFS (months)	PFS6, PFS12	Reference
PARPi monotherapy	OPINION (gBRCA & sBRCA wild-type)	PARPi: 9.2 Placebo: NA	PARPi: PFS12 38.5%; PFS18 24.3%	(Poveda et al., 2022)
	Study 19 (gBRCA & sBRCA wild-type)	PARPi: 7.4 Placebo: 5.5	*PARPi: PFS6 60%; PFS12 30% *Placebo: PFS6 30%; PFS12 10%	(Ledermann et al., 2012)
	NOVA (gBRCA wild-type)	PARPi: 9.3 Placebo: 3.9	*PARPi: PFS6 60%; PFS12 30% *Placebo: PFS6 30%; PFS12 10%	(Mirza et al., 2016)
Bevacizumab monotherapy	**GOG 213 (Paclitaxel + Carboplatin + Bevacizumab followed by Paclitaxel + Carboplatin)	Bev: 13.8 Placebo: 10.4	*Bev: PFS12 (equiv. to PFS6) 60%; PFS18 (equiv. to PFS12) 35% *Placebo: PFS12 50%; PFS18 20%	(Coleman et al., 2017)
	**OCEANS (Gemcitabine + Carboplatin + Bevacizumab followed by Gemcitabine + Carboplatin)	Bev: 12.4 Placebo: 8.4	*Bev: PFS12 (equiv. to PFS6) 55%; PFS18 (equiv. to PFS12) 25% *Placebo: PFS12 25%; PFS18 10%	(Aghajanian et al., 2012)
Doublet trials	**ATALANTE (CT + Bevacizumab + Atezolimumab followed by Bevacizumab + Atezolimumab versus CT +	Bev + Atezo: 15.4 Bev: 13.1	BEV + Atezo: PFS12 (equiv. to PFS6) 56%; PFS18 (equiv. to PFS12) 32% BEV: PFS12 (equiv. to PFS6)	(Kurtz, 2022)

	Bevacizumab followed by Bevacizumab)		46%; PFS18 (equiv. to PFS12) 23%
NRG-004 (CT versus olaparib alone versus olaparib + cediranib)	Ola + Cedira: 18.0 Ola: 12.7 CT: 10.5	Ola + Cedira: PFS6 70%; PFS12 40% Ola: PFS6 60%; PFS12 35% CT: PFS6 70%; PFS12 40%	(Liu et al., 2022)

* Based on best approximation from the PFS KM curve

** These trials involve approximately 6 months of chemotherapy period

After revision (Results, line 197 – line 224 in the marked manuscript):

The efficacy of OPEB-01 can be compared to previous studies on currently available monotherapy options, namely PARPi and bevacizumab maintenance trials involving *BRCA* wild-type, platinum-sensitive recurrent ovarian cancer. In the OPINION trial, which investigated olaparib maintenance monotherapy in 279 patients without the germline *BRCA* mutation, the median PFS was 9.2 months (95% CI 7.6–10.9).¹¹ There are two randomized trials involving PARPi maintenance monotherapy: Study 19¹⁹ for olaparib and NOVA⁵ for niraparib maintenance in platinum-sensitive recurrent disease.⁵ In the placebo groups of these two studies, the median PFS was consistently less than 6 months: 5.5 months for Study 19 and 3.9 months for NOVA. In comparison, in the *BRCA* wild-type, PARPi maintenance subgroup, the median PFS was 7.4 months in Study 19 and 5.5 months in NOVA, translating into an absolute benefit of 1.9 months (HR 0.54, 95% CI 0.34–0.85) in Study 19 and 5.4 months (HR 0.45, 95% CI 0.34–0.61) in NOVA for PARPi compared to placebo. Furthermore, based on the PFS curves of these trials, approximately 70% of *BRCA* wild-type patients who received PARPi showed recurrence by 12 months. These findings are in striking contrast with the results from the OPEB-01 study, where the median PFS was not reached and the 12-month PFS rate was 84.4%.

Efficacy can also be compared with historical randomized trials involving bevacizumab maintenance, namely GOG-213 and OCEANS trial.^{12,13,20} These trials involved a bevacizumab period of approximately 6–7 months prior to maintenance with bevacizumab alone, and thus this chemotherapy time window needs to be factored in when comparing the efficacy outcome with the OPEB-01 study. From the start of maintenance with bevacizumab without chemotherapy, the median PFS would be approximately 5 months for OCEANS.¹³ In GOG-213, which included considerably more patients receiving secondary cytoreductive surgery than OCEANS (35.6% as compared to 11.2%), the median PFS was still approximately 6 months.¹² Therefore, our study with triplet maintenance therapy showed a durable response which—despite being an indirect comparison—exceeded that of previous reports for all currently available monotherapy options for *BRCA* wild-type ovarian cancer with platinum-sensitive recurrence.

After revision (Results, line 229 – line 235 in the marked manuscript): In the ATLANTE trial, with a chemotherapy lead-time was similar to the bevacizumab trials, the median PFS in the bevacizumab plus atezolizumab group was approximately 8.4 months.²² Overall, compared to monotherapy or doublet trials, the outcomes of our study suggest a potential synergy among the three different agents with a notable extension of the median PFS in a recurrent *BRCA* wild-type non-mutated cohort beyond the benchmark of 19.1 months for patients with germline *BRCA* mutations in the SOLO-2 trial.²³

After revision (Limitation, line 272 – 287 in the marked manuscript): Our study was limited by the fact that it was a single-arm, open-label study with a relatively small patient population and no comparator group. The 6-month PFS rate was chosen as the primary endpoint because this was a single-arm phase II study that evaluated signals for quick decision-making; based on previous randomized PARPi monotherapy trials, we expected that a majority of the patients would show recurrence within 6 months without maintenance therapy. Since the 6-month PFS is not a validated surrogate for the overall PFS benefit, further survival maturation is necessary to confirm whether the signs of durable response translate into an OS outcome. Another limitation of our study is the small sample size, which was especially limiting for subgroup analysis of PFS concerning HRD or PD-L1 status. In this regard, a future randomized trial with triplet maintenance may be necessary. With these limitations in mind, the subgroup analysis was particularly limited due to the sample size. Although we were able to assess our primary endpoint, the 6-month PFS rate, the OS data are yet to be matured and will require further investigation. The strength of our study is the homogenous patient population in a platinum-sensitive recurrent setting. All patients were screened for germline and somatic *BRCA* status prior to enrolment. Pre-specified biomarkers, including HRD and PD-L1 status, were also assessed in most patients.

Q3) the important conclusion is the triplete is safe and well tolerated by the patients with majority of grade 1 and 2. However, it will be relevant for the reader to see the % of dose discontinuation individually and all by patients.

A3) Thank you for your valuable comment. We agree with you that adequately describing the safety profile of the triplet maintenance is important. In our study, no patient was permanently discontinued from the study due to toxicity. One patient permanently discontinued bevacizumab, but as per study protocol this patient continued with pembrolizumab and olaparib. Thus, all cases of discontinuation were temporary (i.e., dose interruptions).

We provided the data on dose interruption for each drug by the number of patients in the result section: there were dose interruptions in 29 patients for olaparib, 31 patients for pembrolizumab and 32 patients for bevacizumab. We have also added a summary statistics for AEs as a supplementary table.

After revision (Results, line 121 – 122 in the marked manuscript): The summary statistics for AE is shown in Supplementary Table 1.

After revision: **Supplementary Table 1. Summary statistics for AEs.**

	Patients (N, %)
All grade AEs	44 (100%)
Grade 3 or above	20 (29.5%)
Serous AEs	6 (13.6%)
AEs leading to death	0
AEs leading to dose reduction for olaparib	27 (61.4%)
AEs leading to dose interruption for olaparib	29 (65.9%)
AEs leading to dose interruption for pembrolizumab	31 (70.5%)
AEs leading to dose interruption for bevacizumab	32 (72.7%)
AEs leading to dose interruption for any of the study drugs	36 (81.8%)

After revision (Results, line 127 – 129 in the marked manuscript): With respect to each drug, dose interruptions were required in 29 patients (65.9%) for olaparib, 31 patients (70.5%) for pembrolizumab, and 32 patients (72.7%) for bevacizumab.

After revision (Result, line 133 – 135 in the marked manuscript): One patient permanently discontinued taking bevacizumab due to a side effect, and continued the study with pembrolizumab and olaparib as per the study protocol.

Q4) from the exploratory analysis, the subgroups of CPS1 seems encouraging and need to develop looking to what we are waiting for in the 1st line setting considering several phase III trial on going
A4) We agree with you that the CPS subgroup analysis is notable, but as we mentioned in the limitation section the subgroup analysis was limited due to the sample size. We also look forward to the data from the 1st line phase III trials.

Minor aspects can be also discussed

Q5) about nomenclature BRCA non mutated can be replace by BRCAwt.

A5) Thank you for the recommendation. We agree with you that “BRCA wild-type” may be a more appropriate term for our cohort and made modifications throughout the manuscript.

Q6) other comment figure 3, as there is no correlation between efficacy and previous therapy or PFI, i will be in favor to present the data by HRD score and PDL1 expression

A6) Thank you for your feedback. In particular, we wanted to emphasize that there is a lack of correlation between efficacy and previous therapy/PFI. We would expect poor efficacy in patients with short PFI and vice versa. However, in our study, even those patients with short PFI were disease-free with triplet maintenance for a considerable amount of time (for quite many patients, time on the triplet maintenance is exceeding the PFI after the first-line chemotherapy). To emphasize this, we have ordered Fig. 3 based on the sum of duration of previous therapy leading up to the start of triplet maintenance.

In terms of the stratification factor, the PFS did not differ based on HRD, but did differ based on PD-L1 expression. Based on your feedback, we have created four sub-plots (stratification by HRD and PD-L1 status), and added them as Supplementary Fig. 2.

Also, several modifications were made to Fig. 3 based on the comments from Reviewer #3. There was one patient who voluntarily withdrew from our study, which had not been reflected in our outcome analysis. To correct this error, we have modified the size of the orange bar (triplet maintenance duration) and removed the ongoing arrow for this patient. The KM curves were also modified to reflect this. As for further minor modifications for Fig. 3, we have moved the beginning of triplet maintenance to x=0, added the death mark for one patient with event, and added a vertical dashed line at 6 months so that the primary endpoint of 6 months PFS can be directly read from the figure.

After revision: Fig. 3

After revision: Supplementary Fig. 2. Therapy outcomes stratified by PD-L1 and HRD status.

Q7) adding the median follow up can be help full to interpret the PFS.

A7) The median follow-up duration was 14.0 months (IQR: 8.4 – 17.2) as shown in result section (line 99).

Reviewer #2 – Ovarian cancer clinical trials (Remarks to the Author):

Key results:

This is a single arm phase 2, 44 patient study investigating the maintenance triplet therapy of Olaparib, Pembrolizumab and Bevacizumab in recurrent non-BRCA mutant platinum sensitive ovarian cancer following response to platinum-based chemotherapy. The study reports on the primary efficacy endpoint of PFS at 6 months and safety. The key findings are that this triplet maintenance therapy is safe and shows promising efficacy as evidenced by the PFS6 and perhaps the more meaningful secondary endpoints of PFS12 and median PFS not reached at data cut.

Validity

Q1) The use of the PFS6 as primary endpoint is questionable. PFS6 is not a validated endpoint recognised as a surrogate of overall PFS benefit. This is important here as especially given this group of patients have just responded well to chemotherapy so a high PFS6 rate would be expected. However the high PFS12 months is impressive and supports the authors claims of duration of efficacy.

The authors also recognise the limitations of a single arm study.

A1) Thank you for your feedback.

We acknowledge that PFS6 is not a validated surrogate for the overall PFS benefit. We have chosen 6-months PFS rate as the primary endpoint for the following reasons. First, this is single-arm phase 2 study

which explores whether triplet maintenance has efficacy with tolerable safety. Therefore, we needed a measure for quick decision on whether to proceed to the next step. Second, according to the previous studies, among patients with platinum-sensitive recurrence and CR/PR to the last platinum chemotherapy, more than half of those without maintenance progressed in less than 6 months; in case of PARPi monotherapy, in Study 19, the median PFS was still around 7 months. We have summarized the outcomes from the relevant trials in the summary table below.

We agree with you that the PFS12 of 84.4% from OPEB-01 is encouraging, suggesting a durable response, especially considering that PFS12 was approximately 30% in both Study 19 and NOVA trial (Ledermann et al., 2012; Mirza et al., 2016).

As the reviewer comments, acknowledging the inherent limitation as a single arm phase II trial, we think that an indirect comparison can still be made with the previously published trials involving PARPi monotherapy or bevacizumab in *BRCA* wild-type patients with platinum-sensitive recurrence (please also see Reviewer #1 Q2/A2). The series of outcomes in our study (median PFS, PFS6, and PFS12) are still notable. We have discussed the strength and limitation of our study in the discussion section.

Study type	Trial name	Median PFS (months)	PFS6, PFS12	Reference
PARPi monotherapy	OPINION (gBRCA & sBRCA wild-type)	PARPi: 9.2 Placebo: NA	PARPi: PFS12 38.5%; PFS18 24.3%	(Poveda et al., 2022)
	Study 19 (gBRCA & sBRCA wild-type)	PARPi: 7.4 Placebo: 5.5	*PARPi: PFS6 60%; PFS12 30% *Placebo: PFS6 30%; PFS12 10%	(Ledermann et al., 2012)
	NOVA (gBRCA wild-type)	PARPi: 9.3 Placebo: 3.9	*PARPi: PFS6 60%; PFS12 30% *Placebo: PFS6 30%; PFS12 10%	(Mirza et al., 2016)
Bevacizumab monotherapy	**GOG 213 (Paclitaxel + Carboplatin + Bevacizumab followed by Paclitaxel + Carboplatin)	Bev: 13.8 Placebo: 10.4	*Bev: PFS12 (equiv. to PFS6) 60%; PFS18 (equiv. to PFS12) 35% *Placebo: PFS12 50%; PFS18 20%	(Coleman et al., 2017)
	**OCEANS (Gemcitabine + Carboplatin + Bevacizumab followed by	Bev: 12.4 Placebo: 8.4	*Bev: PFS12 (equiv. to PFS6) 55%; PFS18 (equiv. to PFS12) 25%	(Aghajanian et al., 2012)

	Gemcitabine + Carboplatin)		*Placebo: PFS12 25%; PFS18 10%	
Doublet trials	**ATALANTE (CT + Bevacizumab + Atezolimumab followed by Bevacizumab + Atezolumumab versus CT + Bevacizumab followed by Bevacizumab)	Bev + Atezo: 15.4 Bev: 13.1	BEV + Atezo: PFS12 (equiv. to PFS6) 56%; PFS18 (equiv. to PFS12) 32% BEV: PFS12 (equiv. to PFS6) 46%; PFS18 (equiv. to PFS12) 23%	(Kurtz, 2022)
	NRG-004 (CT versus olaparib alone versus olaparib + cediranib)	Ola + Cedira: 18.0 Ola: 12.7 CT: 10.5	Ola + Cedira: PFS6 70%; PFS12 40% Ola: PFS6 60%; PFS12 35% CT: PFS6 70%; PFS12 40%	(Liu et al., 2022)

* Based on best approximation from the PFS KM curve

** These trials involve approximately 6 months of chemotherapy period

After revision (Results, line 197 – line 224 in the marked manuscript):

The efficacy of OPEB-01 can be compared to previous studies on currently available monotherapy options, namely PARPi and bevacizumab maintenance trials involving *BRCA* wild-type, platinum-sensitive recurrent ovarian cancer. In the OPINION trial, which investigated olaparib maintenance monotherapy in 279 patients without the germline *BRCA* mutation, the median PFS was 9.2 months (95% CI 7.6–10.9).¹¹ There are two randomized trials involving PARPi maintenance monotherapy: Study 19¹⁹ for olaparib and NOVA⁵ for niraparib maintenance in platinum-sensitive recurrent disease.⁵ In the placebo groups of these two studies, the median PFS was consistently less than 6 months: 5.5 months for Study 19 and 3.9 months for NOVA. In comparison, in the *BRCA* wild-type, PARPi maintenance subgroup, the median PFS was 7.4 months in Study 19 and 5.5 months in NOVA, translating into an absolute benefit of 1.9 months (HR 0.54, 95% CI 0.34–0.85) in Study 19 and 5.4 months (HR 0.45, 95% CI 0.34–0.61) in NOVA for PARPi compared to placebo. Furthermore, based on the PFS curves of these trials, approximately 70% of *BRCA* wild-type patients who received PARPi showed recurrence by 12 months. These findings are in striking contrast with the results from the OPEB-01 study, where the median PFS was not reached and the 12-month PFS rate was 84.4%.

Efficacy can also be compared with historical randomized trials involving bevacizumab maintenance, namely GOG-213 and OCEANS trial.^{12,13,20} These trials involved a bevacizumab period of approximately 6–7 months prior to maintenance with bevacizumab alone, and thus this chemotherapy time window needs to be factored in when comparing the efficacy outcome with the OPEB-01 study. From the

start of maintenance with bevacizumab without chemotherapy, the median PFS would be approximately 5 months for OCEANS.¹³ In GOG-213, which included considerably more patients receiving secondary cytoreductive surgery than OCEANS (35.6% as compared to 11.2%), the median PFS was still approximately 6 months.¹² Therefore, our study with triplet maintenance therapy showed a durable response which—despite being an indirect comparison—exceeded that of previous reports for all currently available monotherapy options for *BRCA* wild-type ovarian cancer with platinum-sensitive recurrence.

After revision (Results, line 229 – line 235 in the marked manuscript): In the ATLANTE trial, with a chemotherapy lead-time was similar to the bevacizumab trials, the median PFS in the bevacizumab plus atezolizumab group was approximately 8.4 months.²² Overall, compared to monotherapy or doublet trials, the outcomes of our study suggest a potential synergy among the three different agents with a notable extension of the median PFS in a recurrent *BRCA* wild-type non-mutated cohort beyond the benchmark of 19.1 months for patients with germline *BRCA* mutations in the SOLO-2 trial.²³

After revision (Limitation, line 272 – 287 in the marked manuscript): Our study was limited by the fact that it was a single-arm, open-label study with a relatively small patient population and no comparator group. The 6-month PFS rate was chosen as the primary endpoint because this was a single-arm phase II study that evaluated signals for quick decision-making; based on previous randomized PARPi monotherapy trials, we expected that a majority of the patients would show recurrence within 6 months without maintenance therapy. Since the 6-month PFS is not a validated surrogate for the overall PFS benefit, further survival maturation is necessary to confirm whether the signs of durable response translate into an OS outcome. Another limitation of our study is the small sample size, which was especially limiting for subgroup analysis of PFS concerning HRD or PD-L1 status. In this regard, a future randomized trial with triplet maintenance may be necessary. With these limitations in mind, the Subgroup analysis was particularly limited due to the sample size. Although we were able to assess our primary endpoint, the 6-month PFS rate, the OS data are yet to be matured and will require further investigation. The strength of our study is the homogenous patient population in a platinum-sensitive recurrent setting. All patients were screened for germline and somatic *BRCA* status prior to enrolment. Pre-specified biomarkers, including HRD and PD-L1 status, were also assessed in most patients.

Significance

Q2) This is novel and the first triplet maintenance trial to report out so there is no conflicting published literature. It does align with other reports such as the MEDIOLA trial which has shown that this triplet

non-chemotherapy sparing treatment is effective as a treatment in itself in the platinum sensitive setting, so the promising activity as maintenance treatment is not surprising to me as an expert in this field.

A2) We agree with you on the promising activity of the triplet regimen, which was also suggested in the MEDIOLA trial. Based on the previous reports at the ESMO conference (Banerjee, 2022), the median PFS from the MEDIOLA trial (*gBRCA* wild-type cohort) was 15 months, whereas in our cohort (wild-type for both germline and somatic *BRCA*) the median PFS was not reached. As mentioned in the discussion section, we infer that the triplet combination is potentially more effective as maintenance when started at low tumor burden. From the AE-perspective, our method of starting triplet regimen after chemotherapy was safe and tolerable.

Data and methodology

Methodology –

I would question the inclusion/exclusion criteria in the trial and request more is added to the main manuscript for example:

Q3) 1. Why include non-high grade serous histologies here? There is no data to support this and we know clear cell cancers and low grade serous are totally different cancers with different genetic drivers. The authors should justify this or not.

A3) Since this study was hypothesis generating, we have enrolled patients regardless of histology; yet we put a cap on the maximum number of patients that could be included for non-high grade serous histology (maximum of 8 patients with clear cell carcinoma, protocol p13). In clear cell carcinoma, previous case report of one patient with clear cell carcinoma who received triplet combination (olaparib, pembrolizumab, and bevacizumab) as 3L therapy showed a sustained PR of 12 months (Zhao & Jiang, 2022). In our study, one patient with clear cell carcinoma was enrolled with PR after 2L chemotherapy, and this patient was ongoing at 16 months with no evidence of disease at DCO. Another patient with low grade serous carcinoma was also ongoing with 13 months PFS at DCO. We have added more details regarding histology, which were in the study protocol, to the inclusion criteria in the methods section.

After revision (Methods, line 303 – 307 in the marked manuscript): With respect to histology, patients with high-grade predominantly serous, endometrioid, carcinosarcoma, mixed Mullerian with high-grade serous components, clear cell, or low-grade serous ovarian cancer, primary peritoneal cancer, or fallopian tubal cancer were considered. A cap of 8 patients was applied for clear cell carcinoma; mucinous carcinoma could be enrolled.

Q4) 2. Why was Pembrolizumab not given in C1, what was the rationale for this?

A4) Thank you for the inquiry. Several scientific evidence suggests that DNA damaging agents such as PARPi induces immune cell infiltration and PD-L1 upregulation, leading to an enhanced antitumor immunity which can be further enhanced through the combination of immune check point inhibitor (also mentioned in protocol p10). Based on this pre-clinical background, we included the olaparib lead-in period similar to other PARPi+IO studies such as MEDIOLA or AMBITION study (NCT03699449). We have added the rationale, which were in the study protocol, to the procedure section in under methods.

After revision (Methods, line 324 – 328 in the marked manuscript): Unlike olaparib and bevacizumab which were started in cycle 1, pembrolizumab was initiated in cycle 2, based on the preclinical rationale that PARP inhibitors induce immune cell infiltration and PD-L1 upregulation, leading to enhanced antitumor immunity that can be further enhanced through the combination of an immune check point inhibitor.

Q5) 3. The cycle length for pembrolizumab and bevacizumab are not included in the text and should be (they are in a figure only)

A5) We appreciate your feedback. We have added the cycle lengths in the methods section.

After revision (Methods, line 320 – 324 in the marked manuscript):

Patients received triplet maintenance therapy with olaparib (300 mg tablets, orally twice daily) and bevacizumab (15 mg/kg, intravenously), followed by a combination of 300 mg olaparib twice daily (up to two years and longer in case of PR at two years), 200 mg pembrolizumab every 3 weeks (cycles 2 through 35), and 15 mg/kg bevacizumab every 3 weeks intravenously until progression or intolerable toxicity.

Q6) 4. Which HRD biomarker testing was used?

A6) The HRD biomarker testing was performed based on the allele-specific copy number estimation from our whole-exome sequencing data. Based on the methodology by Sztupinski et al (Sztupinski et al., 2018), parameters such as loss of heterozygosity, large-scale transition, and number of telomeric allelic imbalances were used to calculate the HRD score. Our research team has expertise with calculating HRD score based on the whole-exome sequencing data (Kang et al., 2022). Previous studies have also suggested a good correlation between the earlier SNP array-based approach (such as the Myriad test) and the NGS-based method (de Luca et al., 2020; Sztupinski et al., 2018).

Q7) 5. What the requirement for previous chemo cycles ? at least 4 or must be 6?

A7) According to the study protocol, there was no minimum number of 2L chemotherapy cycles, and patients could be enrolled as long as they showed CR/PR to the 2L chemotherapy. The median number of cycle in our study, however, was 6 cycles (range: 4 to 9 cycles).

Q8) Data itself is presented clearly and figures and tables easy to interpret. There is an error in Figure 1. Its says BRCA mutant but should say 'NON-BRCA mutant'

A8) Thank you for catching this typo; we have modified it "BRCA wild-type."

After revision: Fig. 1.

Analytical approach

Q9) The authors should justify why they chose PFS6 as the primary endpoint given my previous comments as this would be an 'easy win'

A9) We chose PFS6 of 70% as the primary endpoint for this single arm, phase II study. As demonstrated in our previous summary table (A1), we have referred to various clinical trials involving PARPi or bevacizumab maintenance. In the PARPi monotherapy maintenance studies, which enrolled patients with CR/PR to the 2L chemotherapy, the median PFS was less than 6 months for the placebo group and slightly over 6 months for the monotherapy maintenance group. Based on the historical control (GOG-213), we used a HR of 0.5 (such as other triple maintenance therapy study, DUO-O), which we think is a relatively stringent cutoff. As we mentioned in A1, we think that our single arm phase II trial is hypothesis generating; the PFS6, PFS12, and median PFS are notable, suggesting a durable response.

Q10) There is limited analysis due to the small sample size but the association with PDL-1 CPS status is interesting and should be further explored.

A10) We agree with you that the CPS subgroup analysis is notable. We should await to see if the differences in the survival outcome based on the PD-L1 specification is maintained in the future large-scale, prospective, randomized trials such as DUO-O, KEYLINK-001, and FIRST.

Q11) Safety analysis – suggest only use CTCAE definitions in the table so avoid multiple terms for possibly the same thing – hyperthyroidism and thyroiditis for example. More information should be provided in the main text about the 1 patient with the small bowel perforation as this is a concerning toxicity of the bevacizumab.

A11) Thank you for your feedback. We think that the AE reporting related to thyroid function is important, especially since pembrolizumab is included in the triplet maintenance. With respect to AE counting, we have used the exact terms as reported by the clinicians. Such approach is similar to that of KEYNOTE100 and MEDIOLA where the irAE terms for thyroid included thyroiditis, hypothyroidism, and hyperthyroidism.

As for the small bowel perforation, this patient was admitted after 7 cycles of triplet maintenance with abdominal pain, and APCT suggested small bowel perforation. At the time of event, the small bowel perforation was determined to be probably related to bevacizumab – thus, a grade 3 and SAE. This patient was conservatively managed with antibiotics. After 3 weeks of initial event, this patient was found with PD and small bowel obstruction, which also suggested a potential association between the AE and disease progression.

After revision (Methods, line 335 – 336 in the marked manuscript): **With respect to AE reporting, we have adhered to the exact terms used by clinicians.**

After revision (Results, line 142 – 146 in the marked manuscript): **No grade 4 events were observed. One patient experienced grade 3 small bowel perforation after 7 cycles of triplet maintenance therapy. At the time of the event, the small bowel perforation was determined to be probably related to bevacizumab. This patient was conservatively managed with antibiotics, and after 3 weeks, was found with PD and small bowel obstruction.**

Suggested improvements

Q12) Suggested improvements detailed in the comments in the above sections

Also mention OCEANs phase 3 maintenance BEV trial as perhaps a better comparator than GOG213 as the later study also included surgery. (Oceans Trial Aghajanian C et al., JCO 2012)

A12) Thank you for your insightful comment. We agree with you that the OCEANS trial, which does not include secondary debulking surgery, is an excellent comparator. We have made the following modifications in the introduction and discussion section.

After revision (Introduction, line 57 – 63 in the marked manuscript): Another approved maintenance option for platinum-sensitive recurrence is bevacizumab, an antiangiogenic agent; however, with GOG-213, the addition of bevacizumab led to a median PFS gain of 3.4 months irrespective of the *BRCA* status. However, the median PFS gain from adding bevacizumab was 3.4 months in GOG-213¹² and 4.0 months in the OCEANS trial.¹³ Outcomes from these historical trials suggest that the use of an antiangiogenic agents as monotherapy may be insufficient for recurrent disease.

After revision (Discussion, line 213 – 224 in the marked manuscript): Efficacy can also be compared with historical randomized trials involving bevacizumab maintenance, namely GOG-213 and OCEANS trial.^{12,13,20} These trials involved a bevacizumab period of approximately 6–7 months prior to maintenance with bevacizumab alone, and thus this chemotherapy time window needs to be factored in when comparing the efficacy outcome with the OPEB-01 study. From the start of maintenance with bevacizumab without chemotherapy, the median PFS would be approximately 5 months for OCEANS.¹³ In GOG-213, which included considerably more patients receiving secondary cytoreductive surgery than OCEANS (35.6% as compared to 11.2%), the median PFS was still approximately 6 months.¹² Therefore, our study with triplet maintenance therapy showed a durable response which—despite being an indirect comparison—exceeded that of previous reports for all currently available monotherapy options for *BRCA* wild-type ovarian cancer with platinum-sensitive recurrence.

Clarity and context

Well written and clear

Error in Figure 1 pointed out in the previous sections.

References

Yes appropriate references used but also add the OCEANS trial reference

Your expertise

Nothing outside of my scope although I do have statistical expertise.

Reviewer #3 – Biostatistics, clinical trials (Remarks to the Author):

Dr. Kim and colleagues presented a multi-center, single-arm phase II study of triplet maintenance in BRCA non-mutated, platinum-sensitive recurrent ovarian cancer. The paper is well written, but some issues about the statistical analysis and swimmer plot need to be addressed.

Q1) 1. The abstract needs more detailed information including the number of patients in the study, the study design (Simon's two stage).

A1) Thank you for the feedback. We have modified the abstract accordingly.

After revision (Abstract, line 27 – 30 in the marked manuscript): ~~This multicenter, open-label, single-arm, phase II study with Simon two-stage optimum design investigates the efficacy and safety of triplet maintenance (olaparib, pembrolizumab, bevacizumab) in BRCA non-mutated patients with platinum sensitive recurrent ovarian cancer.~~ This study investigates the efficacy and safety of triplet maintenance (olaparib, pembrolizumab, bevacizumab) in these patients.

After revision (Abstract, line 34 – 35 in the marked manuscript): ~~We observe a 6-months PFS rate of 88.6%...~~ Among the 44 patients, we observe a 6-months PFS rate of 88.6%...

Q2) 2. Line 95: Please confirm that the median follow-up was estimated using the Kaplan-Meier method. If that was not the case, please modify accordingly. The IQR can also be estimated from the KM curve for follow-up using the 25th and 75th percentiles.

A2) Thank you for your inquiry; the median follow-up estimates and IQRs were calculated from the Kaplan-Meier curve.

Q3) 3. Line 99, 141: The progression is a time to event outcome, so the median time should be estimated through Kaplan-Meier method.

A3) We appreciate your feedback. In this study, the median survival was not reached, so the median survival time could not be obtained. Thus, we presented the survival rate at specific times (6 months and 12 months) as descriptive statistics that correspond to the Kaplan-Meier curves.

Q4) 4. Line 100: Add text to show that this patient died after PD to make it clear

A4) We have made the modification as per your suggestion.

After revision (Results, line 104 – 105 in the marked manuscript): An overall survival (OS) event occurred in one patient, and this patients died after disease progression.

Q5) 5. Line 102: Confidence interval of ORR should be reported given the statistical analysis part in method section

A5) Since our study involves drugs used as maintenance following chemotherapy, ORR was not included as an outcome in our protocol. Several patients were CR after the 2L chemotherapy without measurable disease.

Q6) 6. Line 288: Genomic scarring is defined here but not reported in the manuscript.

A6) Thank you for pointing this out. By genomic scarring, we had wanted to indicate the genomic scar score. Genomic scar score is estimated based on the loss of heterozygosity, large-scale transition, and number of telomeric allelic imbalances, which can then be used to determine the HRD status based on the cutoff of 42. Our research team has experience with HRD based on whole-exome sequencing data (Kang et al., 2022). Results from the genomic scar score or HRD status was then used for post-hoc analysis with respect to PFS (Fig. S1 and S2).

After revision (Methods, line 359 – 360 in the marked manuscript): Based on the genomic scar score and a cutoff of 42, HRD status was determined.

Q7) 7. Line 298,319: Some secondary endpoints and median duration of response were not reported in the paper. It can be either removed or add those outcomes in the result section

A7) We appreciate your suggestion. We agree with you that the items that were eventually not reported in the paper should be removed.

After revision (Results, line 106 – 108 in the marked manuscript): ~~Among the patients who demonstrated PR after second line chemotherapy (N= 33), the objective response rate was 57.6% during triplet maintenance, which included 15 patients (44.5%) achieving CR during the treatment.~~

After revision (Methods, line 365 – 367 in the marked manuscript): The secondary endpoints included PFS, overall survival (OS), and safety, ~~time to progression, time to first subsequent treatment (or death), time to second subsequent treatment, and the time interval between the dates for second and third disease progression.~~

After revision (Methods, line 394 – 395 in the marked manuscript): ~~The median duration of response, PFS, and associated 95% CIs were calculated using the Kaplan–Meier method.~~

Q8) 8. Line 300-301: Censoring date for PFS should be based on last scan date for the patient since the endpoint is using progression per RECIST 1.1. Censoring for OS should be based on actual follow-up for each patient. These dates could be after the September 20, 2022 cut-off but then the correct analysis would be to use last scan date or September 20, 2022, whichever occurred first. Similarly for OS.

A8) Thank you for your observation. Patients in this study were followed up carefully, except for one patient who voluntarily withdrew from the trial. We have evaluated the PFS and OS based on the method you have mentioned, using the last scan date (or DCO date, if the last scan date falls beyond this date). Our statement in the Method section can be misleading, so we have modified it.

After revision (Methods, line 368 – 369 in the marked manuscript): ~~The cut-off date was September 20, 2022, which was used as the censoring date for PFS and OS.~~

Q9) 9. Fig 3:

a. There were 44 patients reported in this plot with 35 patients who are still receiving treatment. This is contradicted with the Fig 1 in which indicated that 33 patients are ongoing at the data cutoff.

A9) Thank you for your observation. As specified in Fig.1, 33 out of 44 patients are on-going without disease progression (excluding 10 patients with PD & 1 patients with voluntary withdrawal). The confusion is caused by one patient who experienced disease progression, yet was continued on maintenance based on clinician's discretion. This one patient is marked in Fig.3, the 7th patient counting from the top, who is marked with PD as well as an arrow for ongoing status. Thus, in Fig. 3. the number of patients who are ongoing (either PD or not PD) would be 34. There was an error of putting an arrow for ongoing status, where one patient with voluntary withdrawal was not reflected. To reflect the correction of this error, we have modified the size of orange bar (triplet maintenance duration) and removed the arrow from this patient.

For the rest of your comments for Fig. 3 (Q10 – Q13), we found them extremely helpful in terms of reflecting our data. We have made modifications by (1) adding the death mark (Q11), (2) moving the start of triplet maintenance to x=0 (Q12), and (3) adding a vertical line at 6 months to visually show 6-month PFS rate (Q13).

Q10) b. Add the death mark to the plot

A10) Please refer to the revised Fig. 3.

Q11) c. Add the first response mark so we can know duration of response

A11) ORR and duration of response was not reported as we stated in A5 and A7, respectively.

Q12) d. Move beginning of triplet maintenance therapy to $x=0$ as this is the primary endpoint of the study.

A12) Please refer to the revised Fig. 3.

Q13) e. Add a vertical line at 6 months so primary endpoint (6-month PFS rate) can be read from the figure.

A13) Please refer to the revised Fig. 3.

References for the revision

Aghajanian, C., Blank, S. V., Goff, B. A., Judson, P. L., Teneriello, M. G., Husain, A., Sovak, M. A., Yi, J., & Nycum, L. R. (2012). OCEANS: a randomized, double-blind, placebo-controlled phase III trial of chemotherapy with or without bevacizumab in patients with platinum-sensitive recurrent

- epithelial ovarian, primary peritoneal, or fallopian tube cancer. *J Clin Oncol*, 30(17), 2039-2045. <https://doi.org/10.1200/JCO.2012.42.0505>
- Banerjee, S. I., M.; Roxburgh, P.; Kim, J.W.; Kim, M.H.; Plummer, R.; Stemmer, S.M.; You, B.; Ferguson, M.; Penson, R.T.; O'Malley, D.M.; Meyer, K.; Gao, H.; Angell, H.K.; Nunes, A.T.; Domchek, S.; Drew, Y. (2022). *Phase II study of olaparib plus durvalumab with or without bevacizumab (MEDIOLA): final analysis of overall survival in patients with non-germline BRCA-mutated platinum-sensitive relapsed ovarian cancer* ESMO congress 2022, Paris, France.
- Coleman, R. L., Brady, M. F., Herzog, T. J., Sabbatini, P., Armstrong, D. K., Walker, J. L., Kim, B. G., Fujiwara, K., Tewari, K. S., O'Malley, D. M., Davidson, S. A., Rubin, S. C., DiSilvestro, P., Basen-Engquist, K., Huang, H., Chan, J. K., Spiertos, N. M., Ashfaq, R., & Mannel, R. S. (2017). Bevacizumab and paclitaxel-carboplatin chemotherapy and secondary cytoreduction in recurrent, platinum-sensitive ovarian cancer (NRG Oncology/Gynecologic Oncology Group study GOG-0213): a multicentre, open-label, randomised, phase 3 trial. *Lancet Oncol*, 18(6), 779-791. [https://doi.org/10.1016/S1470-2045\(17\)30279-6](https://doi.org/10.1016/S1470-2045(17)30279-6)
- Coleman, R. L., Spiertos, N. M., Enserro, D., Herzog, T. J., Sabbatini, P., Armstrong, D. K., Kim, J. W., Park, S. Y., Kim, B. G., Nam, J. H., Fujiwara, K., Walker, J. L., Casey, A. C., Alvarez Secord, A., Rubin, S., Chan, J. K., DiSilvestro, P., Davidson, S. A., Cohn, D. E., . . . Mannel, R. S. (2019). Secondary Surgical Cytoreduction for Recurrent Ovarian Cancer. *N Engl J Med*, 381(20), 1929-1939. <https://doi.org/10.1056/NEJMoa1902626>
- de Luca, X. M., Newell, F., Kazakoff, S. H., Hartel, G., McCart Reed, A. E., Holmes, O., Xu, Q., Wood, S., Leonard, C., Pearson, J. V., Lakhani, S. R., Waddell, N., Nones, K., & Simpson, P. T. (2020). Using whole-genome sequencing data to derive the homologous recombination deficiency scores. *NPJ Breast Cancer*, 6, 33. <https://doi.org/10.1038/s41523-020-0172-0>
- Kang, H. G., Hwangbo, H., Kim, M. J., Kim, S., Lee, E. J., Park, M. J., Kim, J. W., Kim, B. G., Cho, E. H., Chang, S., Lee, J. Y., & Choi, J. K. (2022). Aberrant Transcript Usage Is Associated with Homologous Recombination Deficiency and Predicts Therapeutic Response. *Cancer Res*, 82(1), 142-154. <https://doi.org/10.1158/0008-5472.CAN-21-2023>
- Kurtz, J. E. P.-L., E.; Oaknin, A.; Belin, L.; Tsibulak, I.; Cibula, D.; Vergote, I.; Rosengarten, O.; Rodrigues, M.; de Gregorio, N.; Martinez-Garcia, J.; Pautier, P.; Mouret Reynier, M.A.; Selle, F.; D'Hondt, V.; Joly Lobbedez, F.; Bultot Boissier, E.; Floquet, A.; Heudel, P.-E.; Heitz, F. . (2022). *A randomized, doublet-blinded, phase III study of atezolizumab versus placebo in patients with late relapse of epithelial ovarian, fallopian tube, or peritoneal cancer treated by platinum-based chemotherapy and bevacizumab. GINECO-OV236b/ENGOT-ov29* ESMO congress 2022, Paris, France.
- Ledermann, J., Harter, P., Gourley, C., Friedlander, M., Vergote, I., Rustin, G., Scott, C., Meier, W., Shapira-Frommer, R., Safra, T., Matei, D., Macpherson, E., Watkins, C., Carmichael, J., & Matulonis, U. (2012). Olaparib maintenance therapy in platinum-sensitive relapsed ovarian cancer. *N Engl J Med*, 366(15), 1382-1392. <https://doi.org/10.1056/NEJMoa1105535>
- Liu, J. F., Brady, M. F., Matulonis, U. A., Miller, A., Kohn, E. C., Swisher, E. M., Cella, D., Tew, W. P., Cloven, N. G., Muller, C. Y., Bender, D. P., Moore, R. G., Michelin, D. P., Waggoner, S. E., Geller, M. A., Fujiwara, K., D'Andre, S. D., Carney, M., Alvarez Secord, A., . . . Bookman, M. A. (2022). Olaparib With or Without Cediranib Versus Platinum-Based Chemotherapy in Recurrent Platinum-Sensitive Ovarian Cancer (NRG-GY004): A Randomized, Open-Label, Phase III Trial. *J Clin Oncol*, 40(19), 2138-2147. <https://doi.org/10.1200/JCO.21.02011>
- Mirza, M. R., Monk, B. J., Herrstedt, J., Oza, A. M., Mahner, S., Redondo, A., Fabbro, M., Ledermann, J. A., Lorusso, D., Vergote, I., Ben-Baruch, N. E., Marth, C., Madry, R., Christensen, R. D., Berek, J. S., Dorum, A., Tinker, A. V., du Bois, A., Gonzalez-Martin, A., . . . Investigators, E.-O. N. (2016). Niraparib Maintenance Therapy in Platinum-Sensitive, Recurrent Ovarian Cancer. *N Engl J Med*, 375(22), 2154-2164. <https://doi.org/10.1056/NEJMoa1611310>
- Poveda, A., Lheureux, S., Colombo, N., Cibula, D., Lindemann, K., Weberpals, J., Bjurberg, M., Oaknin, A., Sikorska, M., Gonzalez-Martin, A., Madry, R., Perez, M. J. R., Ledermann, J., Davidson, R.,

- Blakeley, C., Bennett, J., Barnicle, A., & Skof, E. (2022). Olaparib maintenance monotherapy in platinum-sensitive relapsed ovarian cancer patients without a germline BRCA1/BRCA2 mutation: OPINION primary analysis. *Gynecol Oncol*, *164*(3), 498-504.
<https://doi.org/10.1016/j.ygyno.2021.12.025>
- Sztupinski, Z., Diossy, M., Krzystanek, M., Reiniger, L., Csabai, I., Favero, F., Birkbak, N. J., Eklund, A. C., Syed, A., & Szallasi, Z. (2018). Migrating the SNP array-based homologous recombination deficiency measures to next generation sequencing data of breast cancer. *NPJ Breast Cancer*, *4*, 16. <https://doi.org/10.1038/s41523-018-0066-6>
- Zhao, Y., & Jiang, Y. (2022). Remarkable Response to the Triplet Combination of Olaparib with Pembrolizumab and Bevacizumab in the Third-Line Treatment of an Ovarian Clear Cell Carcinoma Patient with an ARID1A Mutation: A Case Report. *Onco Targets Ther*, *15*, 323-328.
<https://doi.org/10.2147/OTT.S362267>

Reviewers' Comments:

Reviewer #1:

Remarks to the Author:

Dear authors many thanks to have answer to the questions already addressed. However, I continue to have some concerns with the hypothesis formulated.

To consider this clinical trial as positive (because of the nature of the single arm study) we need to look at the hypothesis done for the primary endpoint. In this case, 50% PFS6months rate is considered as null and 70% as positive effect. And compared the data to all previous clinical trials already published using maintenance with bev, or bev + atezo or parpi alone. But for the majority as GOG213, Ocean, Atalante, the PFS rate considered at 12 months (equivalent to their PFS6months) is not valid. All these trials start randomization at the beginning of the CT meaning that the PFS rate at 12 months included all patients who progressed during CT period and not taking into account in the OPEB1 trial. If you are considered all these trials starting to evaluate the patients in SD (not only those in PR/CR as in OPEB1) after CT period and look at the probability of PFS rate at 6 months after starting maintenance the rate is between 65% to 75% regarding OCEAN, GOG and ATALANTE data... This need to be re calculate from authors and be more explicit. I would also insist on the fact we are comparing 44 pts very selected to large phase III trial with median follow up of more than 24 months or 36 months if we consider Atalante or study 19...

I agree the PFS rate at 12 months seems to be very encouraging, longer follow up is required to be strong in the analysis and taking into account the mature OS data

Considering the safety data, I appreciated the details reported, we can noticed the high number of pts who need to interrupt the maintenance closed to 66% for olaparib (less than 50% when used alone) and more than 70% for IV therapies (65% with atezo in Atalante but with a median follow up of 36 months)this need to be emphized

Reviewer #2:

Remarks to the Author:

These comments relate to the review of the revised manuscript.

I think the authors have addressed the reviewers comments satisfactorily and the edits/additions to the manuscript are acceptable. Additional details have been added such as details of the patient populations and justifications for allowing non-HGS histology's. The discussion relating to how these results can be interpreted in relation to historical data is now more accurate and reflects the gaps in data knowledge.

I think the manuscript is now suitable for publication pending further editorial review

I have no other comments

Reviewer #3:

Remarks to the Author:

Authors have appropriately responded to all my comments

Reviewer #1 (Remarks to the Author):

Q1. Dear authors many thanks to have answer to the questions already addressed. However, I continue to have some concerns with the hypothesis formulated.

To consider this clinical trial as positive (because of the nature of the single arm study) we need to look at the hypothesis done for the primary endpoint. In this case, 50% PFS6months rate is considered as null and 70% as positive effect. And compared the data to all previous clinical trials already published using maintenance with bev, or bev + atezo or parpi alone. But for the majority as GOG213, Ocean, Atalante, the PFS rate considered at 12 months (equivalent to their PFS6months) is not valid. All these trials start randomization at the beginning of the CT meaning that the PFS rate at 12 months included all patients who progressed during CT period and not taking into account in the OPEB1 trial. If you are considered all these trials starting to evaluate the patients in SD (not only those in PR/CR as in OPEB1) after CT period and look at the probability of PFS rate at 6 months after starting maintenance the rate is between 65% to 75% regarding OCEAN, GOG and ATALANTE data... This need to be re calculate from authors and be more explicit. I would also insist on the fact we are comparing 44 pts very selected to large phase III trial with median follow up of more than 24 months or 36 months if we consider Atalante or study 19...

I agree the PFS rate at 12 months seems to be very encouraging, longer follow up is required to be strong in the analysis and taking into account the mature OS data

A1. Thank you for your feedback on our manuscript. We acknowledge that the 44 patients from OPEB-01 are a selected group, as they demonstrated a response to platinum-based chemotherapy prior to receiving 2L triplet maintenance.

The concern regarding our hypothesis, particularly the use of GOG 213 as a historical comparator, was previously raised by Reviewer #1 and #2. To address this concern, we have drawn a figure below which outlines the study designs of previously reported trials, namely GOG 213 and Study 19, alongside our triplet maintenance study.

We have explained how the study protocol was drawn to calculate the required sample size. At the time of study protocol design, bevacizumab was the only accessible maintenance option for *BRCA* wildtype patients with platinum-sensitive, recurrent ovarian cancer in Korea. And there was no study on bevacizumab maintenance among patients who responded to platinum-based chemotherapy. Given these

circumstances, we had no alternative but to perform statistical calculation based on the best approximation from the previous trials, which are quite unlike our study design.

We have reasons to think that a majority of patients exhibited PR/CR to chemotherapy in these previously reported trials, considering what we expect for a similar cohort of platinum-sensitive, recurrent ovarian cancer. However, it is not feasible to use statistical techniques to infer PFS rate among those who responded to platinum-based chemotherapy based solely on officially published data. And even if we were able to do so, it would not be appropriate to formulate a null hypothesis based on these approximations.

Overall, we think that our assumption of null hypothesis is not unreasonably low, especially when considering that *BRCA* wildtype patients receiving no maintenance or PARPi showed a median PFS of 5-7 months in Study 19.

To address concerns regarding the formulated hypothesis, we have taken the following actions.

First, we acknowledged the differences between GOG213 and OPEB-01, and have explicitly mentioned them in our previous revision (lines 393 – 400).

We understand that there may still be debates regarding the estimated duration of bevacizumab maintenance post-chemotherapy to be approximately 6 months. Thus, we decided to remove the paragraphs where we made indirect comparisons between our study and previous studies involving concurrent therapy followed by maintenance therapy (GOG 213, OCEANS, and ATALNTE) (lines 230 – 248). Instead, we focused our comparisons on studies with maintenance therapy such as Study19, NOVA (2L+ PARPi maintenance) and PAOLA-1 (1L olaparib + bevacizumab maintenance).

Although we cannot change the study assumption at this point, we would like to highlight that our 6-months PFS rate was 88.6% (95% CI 75.4 – 96.2). The result is robust, considering that the lower bound of 95% CI was over 75%, even if we acknowledge that OPEB-01 included only patients who have responded to platinum-based chemotherapy.

We are not to compare our single arm, phase II data to large phase III trials. However, we still believe that our trial provides a strong early signal of the efficacy of triplet maintenance in *BRCA* wildtype, recurrent ovarian cancer. The data from our subgroup analysis, which indicates that the efficacy was particularly pronounced in the HRD positive and PD-L1 positive subgroup compared to the negative subgroup, provides important insights that can help form the hypothesis for future large, phase III trials in *BRCA* wildtype, recurrent ovarian cancer.

Second, in response to the reviewer's comment on the need for longer follow-up, we have conducted a new data cut-off and re-analyzed all the data. The current median follow-up duration is 22.4 months, which we believe is sufficient to provide a robust efficacy signal. Our current follow-up duration is also comparable to the time when the primary analysis was released in PAOLA-1 or DUO-O.

In our triplet maintenance in 2L recurrent setting, the updated median PFS is 22.4 months, with a 12-months PFS rate of 84% and 18 months PFS rate of 71.4%. Our median PFS exceeds the durations reported in previous 2L maintenance (7.4 months in Study 19 and 9.3 months in *BRCA* wildtype subgroup of NOVA), as well as the median PFS observed in 1L maintenance study with a doublet therapy (18.9 months in *tBRCA* wildtype in PAOLA-1). Based on the KM curve approximation, the 12-months PFS rate in the PARPi monotherapy and placebo group was 30% and 10%, respectively, in both Study 19 and NOVA. Compared to these metrics, our 12-months PFS rate is 84.0%, indicating durable responses. Since

all patients, except for one who withdrew from the study, were followed up for at least 15 months, we can clinically interpret our 12-months PFS finding.

We have included the updated PFS curve below. Also, we have included the revisions made to the main manuscript.

After revision (PFS curve):

After revision (discussion, lines 204-229):

The efficacy of OPEB-01 can be compared to previous studies on currently available monotherapy options, namely PARPi and bevacizumab maintenance trials involving *BRCA* wild-type, platinum-sensitive recurrent ovarian cancer. In the OPINION trial, which investigated olaparib maintenance monotherapy in 279 patients without the germline *BRCA* mutation, the median PFS was 9.2 months (95% CI 7.6–10.9).¹¹ There are two randomized trials involving PARPi maintenance monotherapy in *BRCA* wildtype patients: Study 19¹⁹ for olaparib and NOVA⁵ for niraparib maintenance in platinum-sensitive recurrent disease.⁵ In the placebo groups of these two studies, the median PFS was consistently less than 6 months: 5.5 months for Study 19 and 3.9 months for NOVA. In comparison, in the *BRCA* wild-type, PARPi maintenance subgroup, the median PFS was 7.4 months in Study 19 and 5.5^{9.3} months in NOVA, translating into an absolute benefit of 1.9 months (HR 0.54, 95% CI 0.34–0.85) in Study 19 and 5.4 months (HR 0.45, 95% CI 0.34–0.61) in NOVA for PARPi compared to placebo. Furthermore, based on the PFS curves of these trials, the 12-months PFS rates were approximately 30% in PARPi monotherapy group and 10% in placebo group in these two trials. ~~approximately 70% of *BRCA* wild-type patients who received PARPi showed recurrence by 12 months.~~ These findings are in striking contrast with the results from the OPEB-01 study, where the median PFS was not reached and the 12-month PFS rate was 84.484.0%.

Furthermore, our efficacy outcome has surpassed the median PFS of 18.9 months in the somatic *BRCA* wildtype subgroup of the PAOLA-1 study, which was a frontline maintenance study with doublet regimen (olaparib and bevacizumab).¹⁵ Overall, compared to monotherapy or doublet trials, the outcomes of our study suggest a potential synergy among the three different agents with a notable extension of the median PFS in a recurrent *BRCA* wild-type cohort even beyond the benchmark of 19.1 months for patients with germline *BRCA* mutations in the SOLO-2 trial.²⁰ ~~Thus, our data have raised our expectations for the ongoing randomized trials involving PARP inhibitor based triplet combination as maintenance therapy in the frontline setting.~~

After revision (discussion, lines 230-248):

~~Efficacy can also be compared with historical randomized trials involving bevacizumab maintenance, namely GOG 213 and OCEANS trial.^{12,13,21} These trials involved a bevacizumab period of approximately 6–7 months prior to maintenance with bevacizumab alone, and thus this chemotherapy time window needs to be factored in when comparing the efficacy outcome with the OPEB-01 study. From the start of maintenance with bevacizumab without chemotherapy, the median PFS would be approximately 5 months for OCEANS.¹³ In GOG 213, which included considerably more patients receiving secondary cytoreductive surgery than OCEANS (35.6% as compared to 11.2%), the median PFS was still approximately 6 months.¹² Therefore, our study with triplet maintenance therapy showed a durable response which—despite being an indirect comparison—exceeded that of previous reports for all currently available monotherapy options for *BRCA* wild type ovarian cancer with platinum sensitive recurrence.~~

~~Furthermore, the efficacy results of our study were superior to those of a previously reported doublet therapy with olaparib and cediranib in a platinum sensitive recurrent setting in *BRCA* wild type patients. In the NRG-004 study, the median PFS was 8.9 months (95% CI 8.3–11.2) in the olaparib plus cediranib group, which was not improved when compared with that in the chemotherapy group.²² In the ATLANTE trial, with a chemotherapy lead time was similar to the bevacizumab trials, the median PFS in the bevacizumab plus atezolizumab group was approximately 8.4 months.²³~~

Q2. Considering the safety data, I appreciated the details reported, we can noticed the high number of pts who need to interrupt the maintenance closed to 66% for olaparib (less than 50% when used alone) and more than 70% for IV therapies (65% with atezo in Atalante but with a median follow up of 36 months)this need to be emphasized

A2. We agree with you that AE profile is important. We have updated all the AE data with the new data cut-off, and we have added the AE summary table below for your reference.

Supplementary Table 1. Summary statistics for AEs.

	Patients (N, %)
All grade AEs	44 (100%)
Grade 3 or above	23 (52.3%)
Serous AEs	9 (20.5%)
AEs leading to death	0
AEs leading to dose reduction for olaparib	27 (61.4%)
AEs leading to dose interruption	
Any of the study drugs	38 (86.4%)
Olaparib	32 (72.7%)
Pembrolizumab	34 (77.3%)
Bevacizumab	33 (75.0%)
AEs leading to dose discontinuation	
Any of the study drugs	5 (11.4%)
All three study drugs	1 (2.3%)

As you have pointed out, the rate of dose reduction and interruption are high in our study. There could be potential reasons for this.

First, since all patients in our cohort are Asian, there might be ethnic differences. Second, our low discontinuation rate can be attributed to our proactive management of AEs through dose reduction or interruption. Compared to other studies where a significant number of patients were frequently discontinued from any of the study drugs (e.g., 32.3% in MEDIOLA with 31.9 months median follow up,

26% in DUO-O with median follow up of 23.3 months, and 45% in ATALANTE with median follow up of 36.6 months), our discontinuation rate was 11.4%. Indeed, the rates of dose reduction or interruption in our study were higher than those reported in previous studies on doublet therapy. Regarding dose interruption, our rate of 86.4% is higher than the rate of 54% in PAOLA-1 or 65% in ATALANTE . Similarly, in terms of dose reduction, our rate of 61.4% is higher than the rate of 41% in PAOLA-1. However, it is worth noting that the AE statistics in our study may be similar to other triplet studies, such as DUO-O. For example, the rate of AE leading to dose modification was 76% in DUO-O (dose interruption rate was not reported), and AE profiles were comparable to our study. Third, with a follow-up duration of 22.9 months, we still have a significant number of patients who are continuing their treatment, and they continue to contribute to the AE incidence.

Before revision (discussion): The toxicity profile in our study was in line with that of previous studies on mono and doublet therapies. The most common AEs were hematologic toxicities, including anemia (any grade 36.4%; grade ≥ 3 20.5%) and neutropenia (any grade 29.5%; grade ≥ 3 6.8%). Both the toxicity rate and profile were similar to those in previous studies on olaparib monotherapy (anemia of any grade 16.9%–46.0%; grade ≥ 3 5.1%–21.0%; neutropenia of any grade 15.8%–24.0%; grade ≥ 3 1.8%–8.0%).^{11,19,23} Although the rate of immune-mediated AEs (75.0%) was higher in our study than the reported rate of 22.6% in the Keynote 100 study,²⁴ the events were mostly mild (grade 1 or 2). One of the most common immune-related AEs in our study was thyroiditis (20.5%), which was thyroid-related and thus similar to the most common AE in the Keynote 100 study, which was hypothyroidism (10.1%). Overall, the AEs and immune-related AEs were in line with those observed previously in the respective monotherapy studies, showing no evidence of drug–drug interactions among the three agents.

After revision (discussion, lines 261-274): In terms of AE-related statistics, our study had high dose reduction and interruption rates, 61.4% and 86.4% (for any of the three study drug), respectively. These rates were higher than those reported in previous studies on doublet regimen. For instance, our dose interruption rate of 86.4% surpassed the 54% in PAOLA-1¹⁵ or 65% in ATALANTE.²³ Similarly, our dose reduction rate of 61.4% was also higher than the 41% observed in PAOLA-1. There could be potential reasons. First, since all patients in our cohort are Asian, there may be ethnic differences. Second, we managed to achieve a low discontinuation rate through active dose reduction or interruption. In contrast, other studies frequently experienced discontinuation of the study drugs, such as 32.3% in MEDIOLA¹⁸ with 31.9 months median follow up and 26% in DUO-O²⁵ with median follow up of 23.3 months, whereas our study observed a discontinuation rate of 11.4%. Third, the triplet regimen may have higher toxicity compared to mono or doublet regimen. However, the safety profile was generally consistent with that of the previous triplet regimen (DUO-O).²⁵ The rate of AEs leading to dose modification was 76% in DUO-O (dose interruption rate was not reported), and our AE profiles were also similar.

Reviewers' Comments:

Reviewer #1:

Remarks to the Author:

I would like to warmly thanks the authors to have updated the data, answered to all questions and give more follow up to the study able to report better the PFS and OS

I have no other questions to be resolved except a recomandation: as the authors cited the DUO trial for safety , I will encourage them to also mention the activity of the triplet in the 1st line setting and compared to their own trial as they are doing for PAOLA1

Dear Editor and reviewers,

Thank you for reviewing our manuscript and considering it for publication. Please find our point-by-point response below.

Reviewer #1 (Remarks to the Author):

I would like to warmly thanks the authors to have updated the data, answered to all questions and give more follow up to the study able to report better the PFS and OS

I have no other questions to be resolved except a recomandation: as the authors cited the DUO trial for safety , I will encourage them to also mention the activity of the triplet in the 1st line setting and compared to their own trial as they are doing for PAOLA1

A1. Thank you for assisting us in improving our manuscript. We are cautious about comparing the efficacy of DUO-O with ours due to the differences in study design and line of therapy. However, as per the reviewer's comments, we have discussed the efficacy of DUO-O for readers' understating.

After revision (discussion, line 197-209): DUO-O, a randomized, placebo-controlled phase III trial, showed a significant improvement in PFS with first-line chemotherapy with durvalumab and bevacizumab, followed by maintenance durvalumab, bevacizumab, and olaparib compared with control in patients with *BRCA* wild-type ovarian cancer.²² The median PFS in DUO-O in the triplet maintenance arm was 24.2 months from the randomization. Direct comparisons between DUO-O and our study need to be interpreted with caution due to the differences in study design and the line of therapy. However, as shown in our study, the DUO-O study showed efficacy of the triplet combination.